# Descriptive History Representations: Learning Representations by Answering Questions

## Abstract

Effective decision making in partially observable environments requires compressing long interaction histories into informative representations. We introduce *descriptive history representations (DHRs)*: sufficient statistics characterized by their capacity to answer *relevant questions* about past interactions and potential future outcomes. DHRs focus on capturing the information necessary to address task-relevant queries, providing a structured way to summarize a history for optimal control. We propose a learning framework, involving representation, decision, and question-asking components, optimized using a joint objective that balances reward maximization with the representation's ability to answer informative questions. This yields representations that capture the salient historical details and predictive structures needed for effective decision making. We validate our approach on recommendation tasks with public movie and shopping datasets, generating interpretable textual *user profiles* which serve as sufficient statistics for predicting preference-driven behavior of users.

## 1 Introduction

Reinforcement learning (RL) in partially observable environments is challenging: agents must make decisions based on long interaction histories. This is particularly acute in domains like recommender systems, where an agent must infer a user's latent preferences (the unobserved state) from their observed behavior to make personalized decisions (Åström, 1965; Sondik, 1971; Kaelbling et al., 1998). Representing this history in a compact, informative way is crucial. Traditional approaches in partially observable Markov decision processes (POMDPs) use *belief states* (Åström, 1965; Sondik, 1971; Kaelbling et al., 1998), but these often require a pre-specified Markovian state space, which is impractical for complex user modeling. Conversely, directly using the entire history becomes computationally intractable as it grows. *Predictive state representations (PSRs)* (Littman and Sutton, 2001; Boots et al., 2011) address this by representing the state as a set of predictions about future observations, conditioned on specific action sequences.

In this work, we introduce *descriptive history representations (DHRs)*, a framework that learns representations that focus on *answering questions*. Instead of predicting low-level observations, we construct history representations that answer broad classes of predictive questions. These questions, ideally formulated flexibly (e.g., in natural language), aim to capture essential information for effective decision-making. DHRs shift the representation burden away from predicting specific, low-level observations to a higher level of abstraction, ensuring they answer *relevant* queries. For example, consider a recommender agent that interacts with users. A *user profile*, representing a user's past interactions with the agent, should suffice to answer queries such as: "Does the user prefer Item 1 to Item 2?"; "How likely is the user to abandon the session?"; or open queries like "Write a review this user might provide for the following product: <description>." Such queries can be far more natural, task-relevant, and easier to reason about, than the precise sequence of low-level future observations (e.g., specific interactions) that simply correlate with user satisfaction.

To *learn* DHRs, we employ a multi-agent cooperative training paradigm. A *representation (DHR) encoder* learns to construct a summary of the history, while an *answer agent* learns to generate answers to queries about the history / future using *only* the generated summary. Finally, a *decision agent* determines the next action to take given the current DHR summary. The agents' objectives are aligned with expected reward maximization in the underlying environment.

Our key contributions are as follows. (1) We introduce DHRs, defined by their ability to answer relevant questions about past interactions and future outcomes. We formally establish them as *sufficient statistics* (formally defined in Section 2) for effective decision-making. (2) We propose a multi-agent framework for learning DHRs, using representation, decision, and question-asking components, jointly optimized to balance reward maximization with the representation's ability to answer informative questions. (3) We demonstrate the efficacy of the learned DHRs on recommendation domains, using public movie and shopping datasets, showcasing their ability to generate predictive textual user profiles and improve recommendation quality.

## 2 PROBLEM SETTING

We consider a *partially observable environment (POE)* defined by the tuple $(\mathcal{O}, \mathcal{A}, T)$, where $\mathcal{O}$ is the observation space, $\mathcal{A}$ is the action space, and $T : (\mathcal{O} \times \mathcal{A})^* \mapsto \Delta_{\mathcal{O}}$ is the transition function, mapping (action-observation) histories to a distribution over observations. We assume a *reward function* over histories $R : (\mathcal{O} \times \mathcal{A})^* \mapsto \mathbb{R}$. At each time $t$, the environment is in a history state $h_t = (o_1, a_1, \ldots, o_t)$. An agent takes an action $a_t \in \mathcal{A}$, receives reward $r_t = R(h_t, a_t)$, and the environment transitions to a new state via $o_{t+1} \sim T(\cdot|h_t, a_t)$. A *policy* $\pi : (\mathcal{O} \times \mathcal{A})^* \times \mathcal{O} \mapsto \Delta_{\mathcal{A}}$ maps histories to distributions over actions. Let $\Pi$ denote the set of policies. The *value* of policy $\pi \in \Pi$ is its expected cumulative return: $V(\pi) = \mathbb{E}\big[\sum_{t=1}^{H} r(h_t, a_t)|a_t \sim \pi(h_t)\big]$, where $H$ is the horizon. Our objective is to find a policy $\pi^*$ with maximum value, i.e., $\pi^* \in \arg\max_{\pi} V(\pi)$.

We use the notion of an $f$-*sufficient statistic* below. Let $f : \mathcal{H} \times \Pi \mapsto \mathbb{R}$ be a specific function of interest. We say a mapping $E : \mathcal{H} \mapsto \mathcal{Z}$ is an $f$-sufficient statistic of $\mathcal{H}$ if, for any $h, \pi$, there is some $\tilde{f} : \mathcal{Z} \times \Pi \mapsto \mathbb{R}$ such that $f(h; \pi) = \tilde{f}(E(h); \pi)$ (Rémon, 1984). Examples include value-sufficient statistics for POMDPs such as belief state representations (Åström, 1965), and PSRs (Littman and Sutton, 2001). The main task of our work is to learn such a history-to-state mapping that both effectively summarizes historical information and is amenable to sequential decision making.

Given a probability function $y \sim q(y|x)$, we often write $y \sim q_x$. Let $\mathcal{H}_t$ be the set of histories of length $t$, and $\mathcal{H} = \cup_{t=1}^{H} \mathcal{H}_t$. Similarly, $\Omega_t$ is the set of future realizations beginning at time $t$, with $\Omega = \cup_{t=1}^{H} \Omega_t$. Thus $(h_t, \omega_t) \in \mathcal{H} \times \Omega$ such that $h_t = (o_1, a_1, \ldots, o_t)$ and $\omega_t = (a_t, o_{t+1}, \ldots, o_H)$. For any $\pi \in \Pi$, $d^{\pi}(\omega|h)$ is the probability of a future $\omega$ given $\pi$ starting at history state $h$.

## 3 DESCRIPTIVE HISTORY REPRESENTATIONS (DHRS)

This section formally defines DHRs. We first introduce the *Question-Answer-space (QA-space)*, which structures historical information through questions and answers, and then detail how DHRs serve as compact, actionable summaries derived from these spaces.

### 3.1 QA-SPACES

Informally, a *QA-space* comprises questions and answers over histories. For example, in a conversational recommender, useful questions might query latent user preferences (e.g., "which brand does the user prefer?"), with answers being (distributions over) preference attributes (e.g., "the user prefers brand A over B with probability 0.8"). Alternatively, a question could probe user behavior (e.g., "Write a review this user might

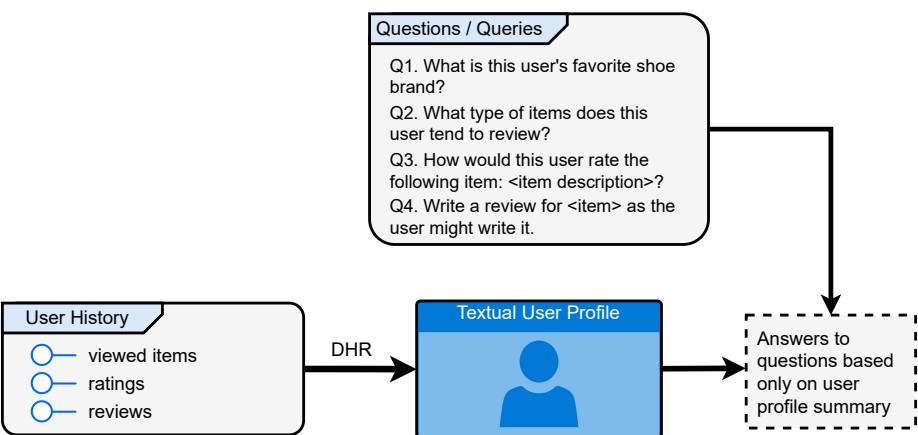

Figure 1: An illustrative example of a descriptive history representation (DHR) which maps a user's history to a compact user profile, which is sufficient for answering questions about the user's preferences.

provide for item A"), answered by a distribution over possible reviews. Formally, a QA-space defines questions and answers which adhere to a particular functional form, where answers are outcomes of questions.

**Definition 1** (QA-space). *A QA-space is a tuple $(\mathcal{Q}, \mathcal{Y}, \mathcal{X}, \nu)$, where $\mathcal{Q}$ is a question space, $\mathcal{Y}$ is an answer space, $\mathcal{X}$ is a context space, and $\nu : \mathcal{X} \times \mathcal{Q} \mapsto \Delta_{\mathcal{Y}}$ is an answer function.*[1]

It is important to note that the set of sufficient questions $Q_x^*$ is determined by the context $x$ (e.g., the history) and the underlying environment dynamics; it exists independently of any specific representation we aim to learn. While QA-spaces are general, in this work we focus on QA-spaces that ask *semantically meaningful* and *interpretable* questions about histories. Thus, the context space $\mathcal{X}$ is the space of histories $\mathcal{H}$, and the answer function $\nu$ poses informative questions about those histories (e.g., summarization of past interactions, or predictions of future events). Notice that the answer function $\nu$ must generally return a *distribution* over the answer space given the partially observable nature of the environment.

We are primarily interested in QA-spaces where the collective answers to questions provide all necessary information for a given purpose (like decision-making or prediction), thereby acting as sufficient statistics. In the recommender setting, a QA-space with user preference questions can serve as a sufficient statistic for future behaviors (e.g., item acceptance, session abandonment) by encoding inferred latent preferences. This motivates the definition of a *sufficient QA-space*, which asks questions that induce a sufficient statistic.

**Definition 2** (*f*-Sufficient QA-space). *A QA-space $(\mathcal{Q}, \mathcal{Y}, \mathcal{X}, \nu)$ is $f$-sufficient, if for any $x \in \mathcal{X}$, there is a subset $Q_x^* \subseteq \mathcal{Q}$ such that $\{(q, \nu(x, q))\}_{q \in Q_x^*}$ is an $f$-sufficient statistic. We call $Q_x^*$ the set of $(f\text{-})$ sufficient questions for context $x$, and $Q^* = \cup_{x \in \mathcal{X}} Q_x^*$ the set of sufficient questions.*

### 3.2 EXAMPLES OF QA-SPACES

**Belief States (Kaelbling et al. (1998)).** For POMDPs with state space $\mathcal{S}$, belief states $b(s|h)$ are distributions over states which constitute a sufficient statistic for value prediction. We define a QA-space for belief states as follows. Let $\mathcal{Q} = \mathcal{S}$, $\mathcal{Y} = [0, 1]$, and answer function $\nu(h, s) = b(s|h)$. For any history $h \in \mathcal{H}$, the set of question-answer pairs $\{(s, b(s|h))\}_{s \in \mathcal{S}}$ constitutes a sufficient QA-space.

---

[1]When $\mathcal{Y}$ is continuous, $\nu$ maps questions and contexts to probability measures on the Borel sets of $\mathcal{Y}$.

**Predictive State Representations (PSRs, Littman and Sutton (2001)).** A PSR is defined by a vector of probabilities over a set of core tests $\big(P(\omega^1|h), \ldots, P(\omega^k|h)\big)$, for any history $h \in \mathcal{H}$. Like belief states, PSRs are sufficient statistics. To construct a QA-space for PSRs, let $\mathcal{Q}$ be the set of tests, $\mathcal{Y} = [0, 1]$ as before, and define an answer function as $\nu(h, \omega) = P(\omega|h)$.

**Item Recommendation (informal).** Consider a user interacting with a recommender, where actions are recommended item slates, and observations are user interactions (e.g., choices, ratings, etc.). The recommender exploits a *user profile* representing user preferences learned from interactions (see Fig. 1). Given a user's history and candidate items, useful questions assess preferences over pairs of items, or the likelihood of accepting recommendations. A sufficient statistic includes answers which are predictive of preferences, behavior, and suffice for optimal recommendations.

### 3.3 DESCRIPTIVE HISTORY REPRESENTATIONS OF QA-SPACES

While QA-spaces offer a powerful mechanism for identifying vital information within a history $h$ by finding answers $\nu(h, q)$ to a set of sufficient questions, they do not inherently define how this distilled information should be structured into an effective representation. We next leverage the insights obtained from QA-spaces to learn a condensed representation, which retains the information required to answer *sufficient* questions.

**Definition 3** (Descriptive History Representation). *Let $(\mathcal{Q}, \mathcal{Y}, \mathcal{H}, \nu)$ be an $f$-sufficient QA-space. An embedding $E : \mathcal{H} \mapsto \mathcal{Z}$ is called a Descriptive History Representation (DHR) if there exists $\nu_A : \mathcal{Z} \times \mathcal{Q} \mapsto \Delta_{\mathcal{Y}}$ such that $\nu_A(z, q) = \nu(h, q)$, for any $h \in \mathcal{H}$, where $z = E(h)$, and any sufficient question $q \in Q_h^*$.*

A DHR encapsulates the essence of the history for question answering by learning a representation $E : \mathcal{H} \mapsto \mathcal{Z}$ that maps a history $h \in \mathcal{H}$ to a compact representation $z \in \mathcal{Z}$. In general, $\mathcal{Z}$ can be arbitrary; e.g., if $\mathcal{Z} \subseteq \mathbb{R}^d$, then $E$ is a classic embedding. In our recommendation example, $z \in \mathcal{Z}$ could be a *textual user profile*, describing a user's preferences in natural language (see Fig. 1 for an illustration). The central idea of a DHR is that the representation acts as a proxy for the history, preserving precisely the information required to answer the relevant questions defined by the QA-space. Thus, instead of computing $\nu(h, q)$ from the history $h$, a *compressed* answer function $\nu_A : \mathcal{Z} \times Q \to \Delta_Y$ produces the same answer distribution using only $z = E(h)$. A DHR, denoted by $z = E(h)$, is a powerful concept because it effectively compresses a potentially complex history $h$ into a more compact form $z$. By preserving the means to answer sufficient questions, the DHR encapsulates all task-relevant information defined by the QA-space, suggesting that it is a sufficient statistics of a given task. Indeed, the following theorem establishes this connection (see Appendix E for proof).

**Theorem 1.** *Let $E : \mathcal{H} \mapsto \mathcal{Z}$ be a DHR (Definition 3). Then it is also an $f$-sufficient statistic.*

## 4 LEARNING DESCRIPTIVE HISTORY REPRESENTATIONS

Our goal is to learn a DHR that is effective for downstream decision making. Specifically, we learn a *DHR policy* $\pi = (E, \pi_D)$ consisting of two components: a *DHR embedding* $E : \mathcal{H} \mapsto \Delta_{\mathcal{Z}}$ (i.e., a history *encoder*); and a *decision policy* $\pi_D : \mathcal{Z} \mapsto \Delta_{\mathcal{A}}$ over summarized histories. The DHR policy induces a policy over histories in the obvious way: $\pi(h) = \mathbb{E}_{z \sim E(h)}[\pi_D(z)]$. To ensure $E$ is indeed a DHR (i.e., a sufficient statistic), we train an *answer function* $\nu_A : \mathcal{Z} \times \mathcal{Q} \mapsto \mathcal{Y}$ for any sufficient question $q \in Q_h^*$ (see Definition 3).

**Generating Sufficient Questions and Answers.** To train $\nu_A$, we need to define the set of sufficient questions $Q^*$. We introduce an oracle *QA-generator*—only required during training—which provides ground-truth question-answer pairs $(q, y)$ that define the content used to train the DHR. This follows the paradigm of Learning Using Privileged Information (LUPI) (Vapnik et al., 2015), where extra information is available during training but not at test time. To obtain a QA-generator, we use full trajectory realizations; i.e., a trajectory $\tau := (h, \omega)$, where $h$ is the history, and $\omega$ is its future realization. The QA generator leverages both

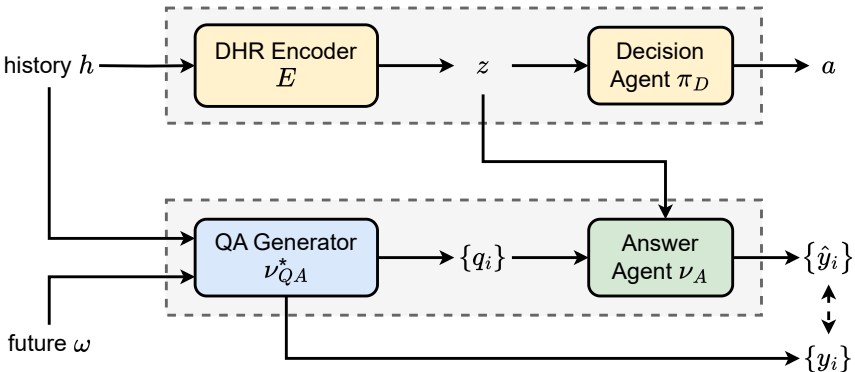

Figure 2: An illustration of our learning framework. A policy $\pi$ is composed of a DHR encoder $E$ and a decision policy $\pi_D$. The DHR embedding takes a history $h$ and creates a representation $z$. The decision policy takes actions in the environment given the representation $z$. In order to learn a DHR ($E$), we use a QA generator, responsible for generating history ($h$) and future ($\omega$)-dependent question-answer pairs. Finally, an answer agent is responsible for answering said questions with access to only the representation $z$. The training procedure is described in Sec. 4 and Algorithm 1.

$h$ and $\omega$ (the privileged information) to construct questions and their corresponding ground-truth answers. Designing this oracle can be challenging. In Sec. 5, we detail how we design a QA-generator using LLMs and future user interactions for recommendation tasks. We refer to Appendix B for a detailed discussion on the construction and training of a QA-generator. Formally, a QA generator is a mapping $\nu_{QA}^* : \mathcal{H} \times \Omega \mapsto \Delta_{\mathcal{Q} \times \mathcal{Y}}$, which we decompose as $\nu_{QA}^*(h, \omega) = \nu_A^*(y|q, h, \omega)\nu_Q^*(q|h, \omega)$.

**Training Objective.** We define the joint objective of the DHR encoder, answer agent, and decision agent to maximize value and match the distribution of ground-truth answers (i.e., match $\nu_A^*$):

$$\max_{E, \nu_A, \pi_D} (1 - \lambda)V(\pi) - \lambda D_f\left(d^{\nu_A^*} \middle\| d^{\nu_A}\right), \tag{OPT 1}$$

where $\lambda \in [0, 1]$ balances between RL and DHR learning, and the distributions are defined as $d^{\nu_A}(y, h, q) = P\big(y, h, q \mid q \sim \nu_Q^*(h, \omega), y \sim \nu_A(z, q), z \sim E(h), \omega \sim d^\pi(\omega \mid h)\big)$, and $d^{\nu_A^*}(y, h, q) = P\big(y, h, q \mid q, y \sim \nu_{QA}^*(h, \omega), \omega \sim d^\pi(\omega \mid h)\big)$. Here, $D_f$ denotes a specific $f$-divergence. Directly optimizing Eq. (OPT 1) is challenging, so we solve its variational form. Let $f^*$ be the convex conjugate of $f$, then the dual formulation of Eq. (OPT 1) is:

$$\max_{\pi_D, E, \nu_A} \min_{g: \mathcal{H} \times \mathcal{Q} \times \mathcal{Y} \mapsto \mathbb{R}} \mathbb{E}\Big[(1 - \lambda)r(h, a) + \lambda\mathbb{E}_{y \sim \nu_A(q, h, \omega)}[f^*(g(h, q, y))] - \lambda\mathbb{E}_{y \sim \nu_A^*(z, q)}[g(h, q, y)]\Big].$$
$$\tag{OPT 2}$$

This max-min objective can be solved by iteratively training a discriminator $g$ and the agents ($\pi_D, E, \nu_A$). Note that when the QA-generator $\nu_{QA}^*$ is fixed (our primary setting), the target distribution $d^{\nu_A^*}$ is stationary. The optimization in Eq. (OPT 2) thus becomes a stable density ratio estimation task, avoiding the instability common in adversarial methods (e.g., GANs) where the target distribution is moving.

**DHR Learning (DHRL).** We now detail the DHRL algorithm (Algorithm 1). The framework is depicted in Fig. 2. DHRL trains the DHR encoder, answer agent, decision agent, and discriminator, with parameters $\theta_E, \theta_A, \theta_D, \theta_g$. It first samples trajectories using $E$ and $\pi_D$ (line 3). These are used in hindsight: each

---

**Algorithm 1** Descriptive History Representation Learning (DHRL)

---

1: Initialize $\theta_E, \theta_A, \theta_D, \theta_g$.
2: **for** each training iteration **do**
3:     Sample trajectory $\tau = (o_1, a_1, \ldots, o_H)$ and rewards $r_1, \ldots, r_H$.                    ▷ Using $E, \pi_D$
4:     **for** $t = 1, \ldots, H$ **do**
5:         Split trajectory to history $h_t$ and future $\omega_t$.
6:         Sample QA pairs $\{(q_{kt}, y_{kt})\}_{k=1}^{K} \sim \nu_{QA}^*(h_t, \omega_t)$.
7:         Sample representation $z_t \sim E(h_t)$.
8:         Predict answers $\hat{y}_{kt} = \nu_A(z_t, q_{kt})$ for $k = 1 \ldots K$.
9:     **end for**
10:     Update $\theta_g$ according to $\frac{1}{HK} \sum_{t=1}^{H} \sum_{k=1}^{K} \nabla_{\theta_g} \left[ f^*(g_{\theta_g}(h_t, q_{kt}, \hat{y}_{kt})) - g_{\theta_g}(h_t, q_{kt}, y_{kt}) \right]$.
11:     Update $\theta_D, \theta_A, \theta_E$ via RL using rewards $r^D, r^A, r^E$ (See Sec. 4).
12: **end for**

---

trajectory is split into histories $h_t$ and futures $\omega_t$ (line 5). The QA-generator $\nu_{QA}^*$ creates ground-truth pairs $(q_k, y_k)$ (line 6). A representation $z_t \sim E(h_t)$ is sampled (line 7), and the answer agent predicts answers $\hat{y}_k = \nu_A(z_t, q_k)$ (line 8). The collected data is used to update the discriminator $g$ (line 10). The agents are updated using RL (line 11), with reward signals derived from Eq. (OPT 2). Specifically, the decision agent $\pi_D$ uses reward $r^D = r(h, a)$, the answer agent uses reward $r^A = f^*(g(h, q, \hat{y}))$, and the DHR embedding uses reward $r^E = (1 - \lambda)r^D + \lambda r^A$. The DHRL framework is flexible and supports both *online* and *offline* training paradigms. In an online setting, trajectories are actively sampled from the environment using the current policy. In an offline setting, they are simply drawn from a fixed, pre-collected dataset. Crucially, the QA-generator, answer agent, and discriminator are used only during training. During deployment the agent *does not have access* to the future. It simply uses the DHR encoder on the current history $h_t$ to produce the representation $z_t$ for the decision agent. The future trajectory is only required for training, not execution, making the inference process efficient.

## 5 EXPERIMENTS

Our experiments are designed to validate DHRL, focusing on its ability to generate predictive representations that answer relevant questions. We demonstrate that DHRL can produce high-quality DHRs in recommendation domains, where we use DHRs to generate textual user profiles which (1) accurately answer questions about user preferences and future behavior; (2) induce informative, coherent summaries of user history; and (3) effectively support a downstream recommendation task.

### 5.1 EXPERIMENTAL SETUP

**Datasets and Task.** We use two datasets in our experiments: MovieLens 25M (Harper and Konstan, 2015), and Amazon Reviews (Ni et al., 2019) (specifically, the *Clothing, Shoes and Jewelry* category). User histories $h_t$ consist of sequences of observations (item titles, ratings, and, for Amazon, descriptions, prices and reviews) and actions (recommended items). This history serves as input to the encoder $E$, which creates a textual user profile. Future user interactions $\omega_t$ are only used in hindsight to construct the QA pairs for training.

**Question Generation.** For both MovieLens and Amazon, questions $q_k$ compare pairs of held-out movies (e.g., "Rank the following movies based on the user's preferences: [<movie_1>, <movie_2>]"), with the ground-truth answer $y_k$ taken from future user ratings in $\omega_t$. For Amazon Reviews, we augment the question set with review generation queries $q_{review}$ which ask "Write a review for item <item description> as <user> might write it." Here, the ground-truth answer $y_{review}$ is the user's textual review of that item from $\omega_t$.

Table 1: Performance of DHRL with default settings (256-token profile, 5 questions, 10 interactions, TV-divergence) after 1,000 iterations for Gemma V3 4B, 12B, and Gemini 1.5 Flash models. Bold indicates statistically significant improvement over the corresponding baseline (p<0.05). See Appendix D for confidence intervals.

| Model & Method | Prediction Accuracy (w.r.t. GT) | Rec. Reward ($r^D$) | Profile-History Consistency (AI/Human) | Prediction Fidelity (AI/Human) | Review Quality (AI/Human) |
|---|---|---|---|---|---|
| *Task: recommendation using Amazon products user profiles* | | | | | |
| Gemma V3 4B | 0.34 | 0.54 | 3.46 / 3.28 | 0.44 / 0.47 | 0.15 / 0.27 |
| Gemma V3 4B+DHRL | **0.71** | **0.83** | 3.41 / **3.85** | **0.56 / 0.53** | **0.85 / 0.73** |
| Gemma V3 12B | 0.67 | 0.78 | 4.32 / 4.18 | 0.34 / 0.39 | 0.17 / 0.35 |
| Gemma V3 12B+DHRL | **0.75** | **0.84** | **4.39 / 4.37** | **0.66 / 0.61** | **0.83 / 0.65** |
| Gemini 1.5 Flash | 0.69 | 0.80 | 4.41 / 4.30 | 0.38 / 0.41 | 0.17 / 0.38 |
| Gemini 1.5 Flash+DHRL | **0.74** | **0.86** | **4.46 / 4.34** | **0.62 / 0.59** | **0.83 / 0.62** |
| *Task: recommendation using MovieLens user profiles* | | | | | |
| Gemma V3 4B | 0.37 | 0.61 | 3.72 / 3.96 | 0.32 / 0.45 | - |
| Gemma V3 4B+DHRL | **0.74** | **0.79** | **3.84 / 4.24** | **0.68 / 0.55** | - |
| Gemma V3 12B | 0.58 | 0.64 | 4.66 / 4.58 | 0.35 / 0.42 | - |
| Gemma V3 12B+DHRL | **0.84** | **0.93** | 4.62 / **4.71** | **0.76 / 0.66** | - |
| Gemini 1.5 Flash | 0.62 | 0.68 | 4.69 / 4.53 | 0.38 / 0.39 | - |
| Gemini 1.5 Flash+DHRL | **0.82** | **0.92** | **4.74 / 4.70** | **0.77 / 0.62** | - |

**Training Methodology.** We train $E$ and $\nu_A$ via standard policy gradient (PG). We use Gemini 1.5 Flash (Team et al., 2024) and Gemma V3 (4B, 12B) (Team et al., 2025) as our language models. All agents use the same base model and share parameters. Our decision agent $\pi_D$ is fine-tuned to make item recommendations to the user given their generated profile (DHR), receiving a reward based on the user's rating for the item. Unless otherwise stated, we use a Gemma V3 4B model, a 256-token profile limit, $K = 5$ item comparison questions, one review question (Amazon only), a history length of $H = 10$ interactions, and TV-distance.

**Evaluation Metrics.** The generated DHRs (user profiles $z$) and the accuracy of the answer agent $\nu_A$ are evaluated using three main criteria: predictive accuracy, recommendation reward, and human evaluation.

For ranking questions, which compare pairs of items, the *predictive accuracy* is the fraction of pairwise comparisons the agent predicts correctly with respect to ground-truth future user ratings. We also define a *recommendation reward* to assess the utility of generated DHRs for our downstream task. Specifically, given a DHR and a held-out set of unseen items, $\pi_D$ recommends an item. The reward is defined by the user's true rating for this item (normalized to $[0, 1]$).

Finally, we use Gemini 2.5 Pro and 24 human raters[2] to assess three qualitative aspects: (1) *Profile-History Consistency*: Raters are given a user interaction history $h_t$ and its generated textual profile $z_t$. They rate (1-5 scale) how accurately and coherently $z_t$ summarizes the information in $h_t$. (2) *Prediction Fidelity*: Raters are given a generated user profile $z_t$ and asked to predict the user's preferences over held-out item pairs. We report the win-rate between the rater accuracy for a baseline-generated profile vs. its DHRL counterpart. Note that human performance on this task may be inherently limited. (3) *Review Quality* (Amazon only): Raters are shown the profile $z_t$ and two reviews for a held-out item, one from a baseline model and one from the DHRL model. We report the win-rate for the DHRL-generated review, indicating its perceived authenticity.

___
[2]Raters were paid contractors. They received their standard contracted wage, which is above the living wage in their country of employment.

Table 2: Comparison with state-of-the-art recommender system baselines. Our DHRL-enhanced model outperforms both the base LLM and specialized RecSys models. Results significantly outperform baselines (p<0.05).

| Dataset | Model & Method | Prediction Accuracy (w.r.t. GT) | Rec. Reward ($r^D$) |
|---|---|---|---|
| Amazon | Gemma V3 12B (Baseline) | 0.67 | 0.78 |
| | DCLMDB (Huang et al., 2024) | 0.73 | 0.82 |
| | DCRec(Yang et al., 2023) | 0.71 | 0.81 |
| | **Gemma V3 12B + DHRL (ours)** | **0.75** | **0.84** |
| MovieLens | Gemma V3 12B (Baseline) | 0.58 | 0.64 |
| | DCLMDB (Huang et al., 2024) | 0.80 | 0.88 |
| | DCRec(Yang et al., 2023) | 0.78 | 0.85 |
| | **Gemma V3 12B + DHRL (ours)** | **0.84** | **0.93** |

## 5.2 RESULTS

Table 1 summarizes the performance of DHRL under our default configuration. Incorporating DHRL leads to substantial improvements in predictive accuracy for ranking tasks. Furthermore, AI and human evaluators rated the generated profiles favorably for consistency with user history and preference predictive ability, suggesting that textual DHRs encapsulate relevant user information effectively. We find human and AI evaluation results to be highly correlated. The notably higher recommendation rewards attained by DHRL-enhanced models confirm that DHR representations are beneficial for downstream decision-making. For Amazon Reviews, review quality metrics also indicate a strong preference for reviews conditioned on DHR profiles over baseline-generated reviews.

We compare our DHRL framework against state-of-the-art methods for recommendation, including: DCLMDB (Huang et al., 2024) and DCRec (Yang et al., 2023). The "Gemma V3 (Baseline)" uses the same LLM backbone trained on the same data via standard predictive objectives. The results in Table 2 show that while these methods improve upon the base LLM, our DHRL framework consistently achieves superior performance in both prediction accuracy and recommendation reward. This demonstrates that DHRL is not only a general representation learning framework but also a highly competitive method in the recommendation domain.

The impact of history length on reward is shown in Fig. 3 (left). DHRL consistently outperforms the baseline. Notably, optimal performance for DHRL is observed with shorter histories (5-10 interactions). This suggests that relatively concise histories suffice for learning effective DHRs in our benchmark tasks, and excessively long histories may introduce noise. Fig. 3 (middle) shows the impact of DHR profile length. Performance generally improves as the token limit increases from shorter lengths, allowing more detailed information. However, this trend does not continue indefinitely; beyond an optimal length (around 256 tokens), gains diminish. This might stem from challenges in exploiting an overly large representation space. Fig. 3 (right) shows that recommendation reward generally increases with $K$, the number of guiding questions. This emphasizes the importance of using a sufficiently diverse set of questions to guide the representation learning process towards a more robust representation of user preferences.

We further validate two key aspects of the DHRL framework via ablation studies (see Appendix D.1). First, we confirm that the DHR closely approximates a sufficient statistic: providing the decision agent with access to the raw history in addition to the DHR ($\pi_D(z_t, h_t)$) yielded negligible improvement (+0.5% reward) over using the DHR alone ($\pi_D(z_t)$). Second, we demonstrate the necessity of the joint optimization (OPT 1). Training the representation using only the QA objective ($\lambda = 1$) significantly underperformed the full DHRL method, confirming that balancing the QA framework with reward maximization is crucial.

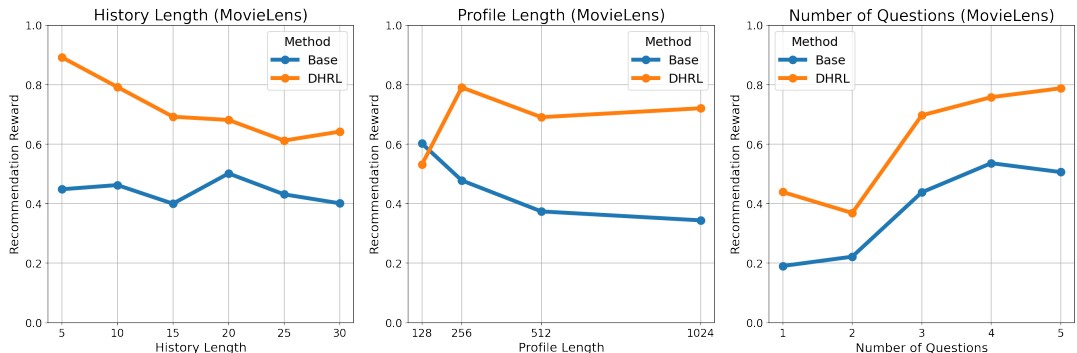

Figure 3: Ablation study on the history length (left), profile length (middle) and number of questions (right) on the MovieLens dataset, showing recommendation reward. DHRL (orange) consistently outperforms the baseline (blue). Short histories (5-10 interactions) are often sufficient. Performance peaks at an intermediate profile length (e.g., 256 tokens). More questions generally improve recommendation quality.

We also experimented with **learning the QA-generator** (see Appendix B). While this led to a marginal improvement in reward (2-3%), the fixed QA-generator bootstrapped with a powerful, pre-trained LLM was already remarkably effective. This highlights the practical applicability of our approach, leveraging off-the-shelf LLMs to create a rich QA-space without complex adversarial training.

Finally, the choice of $f$-divergence significantly impacts performance. Among the tested divergences for the Amazon dataset, TV-distance yielded the best results with a Prediction Accuracy of 0.71 and a Recommendation Reward of 0.83. This substantially outperformed both $\chi^2$-divergence (0.42 accuracy, 0.61 reward) and KL-divergence (0.40 accuracy, 0.58 reward), highlighting that the specific mechanism used to align the answer agent's output distribution with the target distribution is a critical factor in DHRL.

## 6 RELATED WORK

Learning history representations is crucial for POMDPs (Kaelbling et al., 1998). PSRs (Littman and Sutton, 2001) use predictions of future action-observation sequences ("tests"), though as discussed above, rely on manually engineered, often low-level tests. In contrast, DHRs learn representations based on answering high-level, semantically meaningful questions about the future, shifting the focus to task-relevant understanding.

Predictive principles have been adapted to deep learning methodologies. Predictive-state decoders (Venkatraman et al., 2017) and PSRNNs (Downey et al., 2017) predict future observations, features, or latent states. While leveraging prediction, they typically focus on low-level features, in contrast to DHRs, which use a broader, more abstract set of questions (the QA-space) to shape representations, ensuring they answer informative queries beyond standard next-observation prediction. This QA-space can be viewed as learning a set of General Value Functions (GVFs) (Sutton et al., 2011), where questions define the cumulant and policy.

Recent work has emphasized self-prediction in latent space (Ni et al., 2024; Schwarzer et al., 2020), learning representations by predicting future latent states or values. These methods yield latent representations implicitly defined by the prediction objective. DHRs, via the QA framework, explicitly define representation content through their questions, allowing for greater interpretability and control. Other approaches approximate belief states, like the Wasserstein Belief Updater (Avalos et al., 2023), or use frozen pre-trained LLMs for history compression (Paischer et al., 2023). DHRs use a multi-agent framework to learn the representation encoder jointly with answer and decision modules, guided by reward maximization and question-answering accuracy.

Finally, LLMs have been used to generate textual user profiles (Hou et al., 2024; Zhang et al., 2023) to enhance interpretability. DHRs differ as they are directly optimized to serve as sufficient statistics, with information encoded not just by generative summarization, but by their ability to answer task-relevant predictive questions.

## 7 CONCLUSION

This paper introduced descriptive history representations (DHRs), a framework for learning history summaries by focusing on their ability to answer task-relevant questions. We demonstrated its effectiveness on challenging recommendation domains, where our learning approach generates interpretable textual user profiles that act as sufficient statistics. These profiles lead to strong predictive accuracy and outperform powerful baselines in downstream recommendation tasks.

Our work yields two key insights. First, DHRs offer a compelling method for representation learning that enhances interpretability and aligns representations with high-level goals via an explicit QA-space. This provides a structured way to inject task-relevant priors into the learning process, moving beyond simple predictive objectives. Second, we find that modern LLMs are effective at generating semantically rich questions, serving as powerful, off-the-shelf QA-generators, which makes the DHR framework highly practical for a wide range of applications. Future work can explore learning the QA-generator for domains where strong priors are unavailable, and applying the DHR framework to other partially observable domains such as robotics (where questions might resemble GVFs), human-computer interaction, and dialogue systems.

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

## A SOCIETAL IMPACT STATEMENT

Descriptive history representations (DHRs), particularly when viewed as interpretable textual user profiles, offer specific societal benefits but also pose certain risks. We believe the benefits outweigh the risks if specific risk-mitigation strategies are employed in the use of DHRs. On the positive side, this approach can significantly enhance the transparency of user models in partially observable environments like recommender systems. By generating textual summaries that explain the basis for decisions, users might gain a clearer understanding of why certain recommendations are made, fostering trust and potentially allowing for more informed interactions. Moreover, DHRs can support a greater degree of user control and agency. The QA-space itself, guiding the DHR to focus on task-relevant information, could lead to more nuanced and genuinely helpful personalization, as the system learns to explicitly answer questions about user preferences and future behavior rather than just correlations with low-level signals. This explicit focus on question-answering for representation learning offers a pathway to models whose internal reasoning is more transparent.

However, the very interpretability and specificity of DHR-generated textual profiles introduce distinct challenges. Such concentrated, human-readable summaries of user history, even if aimed at better decision-making, could inadvertently reveal sensitive information or enable more precise inference of private attributes if not carefully managed, posing additional privacy risks beyond those of opaque embedding-based profiles. The content and biases of these textual profiles are also directly influenced by the choice of questions in the QA-space and the underlying language models used for generation—an improperly designed QA-space or biased LLM could lead to profiles that amplify societal biases or misrepresent users in an understandable but harmful way. Furthermore, while textual profiles aim for clarity, they are still summaries, and the nuances captured or missed are dictated by the QA-space, potentially leading to over-simplification or misinterpretations that could be exploited if the system is designed to generate persuasive, profile-aligned outputs. Responsible development necessitates careful design of the QA-space to ensure it elicits beneficial and fair representations, alongside ongoing scrutiny of the generated textual profiles for unintended consequences.

# B  QA GENERATOR (DESIGN AND ADVERSARIAL TRAINING)

The descriptive history representation (DHR) framework relies on a question-answer space (QA-space) to guide the learning of informative representations. A crucial component is the QA generator ($\nu_{QA}^*$), which provides question-answer pairs $(q, y)$ based on the history $h$ and future outcomes $\omega$. This section elaborates on how this was handled in our experiments, including an adversarial approach to training the QA generator, and discusses the design of sufficient question sets.

## B.1  EXPERIMENT DESIGN

Our work explores two approaches for the QA generator. For our main experiments, we used a pre-defined QA generator to isolate and evaluate the core components of DHR learning in a controlled setting. This is implemented by procedurally generating questions and their ground-truth answers based on observed user histories and future interactions from the datasets. Additionally, as detailed in Section B.4, we also explored learning the generator adversarially. We found that the practical gains were minor, reinforcing the effectiveness and efficiency of using a powerful, fixed LLM to bootstrap the QA-space.

Our questions are aimed at capturing salient aspects of user preferences and future behavior, rather than predicting raw, low-level observations. In our experiments, ranking questions include comparing pairs of items based on predicted user preference, such as "Does the user prefer Item A or Item B?" The ground-truth answers for these is derived from the user's future ratings of these items.

For the Amazon dataset, we also include review generation questions, asking to generate a textual review for an item as the user might write it; e.g., "Write a review this user might provide the following product: <description>." The ground-truth answer here is the actual review written by the user for that item in their future interactions.

Our questions rely on natural language templates, whose slots are filled with specific items or contexts from the user's future. This approach ensures that questions are not overly specific to individual data points but rather represent generalizable queries about user behavior and preferences, leveraging the richness and flexibility of natural language. This methodology moves beyond attempting to predict the exact sequence of raw future observations (e.g., raw ratings or clicks). Instead, it focuses the representation learning on summarizing history in a way that supports answering these more abstract, semantically meaningful questions, which are directly relevant to the decision-making task, like making good recommendations. Careful design of these questions is crucial for ensuring the DHR learns to encapsulate information relevant to understanding user preferences and predicting their future actions in a structured manner.

## B.2  DESIGNING SUFFICIENT QUESTION SETS

The choice of questions fundamentally defines the information that the DHR aims to capture. A set of questions $Q^*$ is deemed sufficient if the answers to these questions provide all necessary information for a specific purpose, such as maximizing reward in an RL task.

Designing a set of sufficient questions is analogous to other critical design choices in RL, such as defining an action space (or a set of options), an abstraction of the state space, or crafting a reward function. In many RL applications, these components are manually engineered based on domain knowledge and the specific goals of the agent, and are often refined iteratively. For instance, in robotics, actions might be joint torques, and rewards might be based on task completion. Similarly, a set of questions for DHRs can be designed by domain experts to probe the most relevant aspects of the environment's state or history.

A heuristic approach for creating a sufficient question set is to iteratively refine or expand the set of questions based on the agent's performance or by analyzing what information seems to be missing from the DHR. For

example, in a dialogue system, initial questions might focus on user intent or sentiment, with later additions querying user knowledge or specific entities. In autonomous driving, questions could range from simple presence detection like "Is there a pedestrian in the crosswalk?" to more complex predictions like "What is the predicted trajectory of the vehicle ahead?"

A well-designed set of questions can be highly beneficial. Questions, especially in natural language, make the DHR's content more interpretable, as we can understand what information the representation is trying to encode. It also explicitly directs the representation learning process towards capturing information deemed critical by the designer. Furthermore, compared to learning representations based on predicting all raw future observations—as seen in some PSR or belief state approaches—focusing on a curated set of high-level questions can be more tractable and generalizable, particularly in complex environments with high-dimensional observation spaces.

While learning the questions themselves is a desirable long-term goal, designing the question set using the strong prior of large language models (LLMs) is a practical and powerful approach, which injects domain knowledge and task relevance into the representation learning process. This practicality and power stem from the vast world knowledge and semantic understanding embedded within LLMs, acquired during their pre-training on diverse and extensive corpora. Consequently, an LLM can be prompted to generate candidate questions with domain-specific nuances—for example, recognizing relevant attributes for movies, such as genre, director, or thematic elements; or for products, such as brand, material, or user reviews.

Beyond domain specifics, LLMs possess a strong grasp of task-relevant concepts. For a recommendation task, this includes notions of preference, comparison, a user's potential future behavior, or even the underlying reasons for a choice, enabling the generation of questions that directly probe these critical aspects. This allows the LLM-guided design process to formulate questions that elicit more abstract and semantically rich answers than merely predicting low-level future observations. For instance, rather than focusing solely on predicting the next click, questions can probe comparative preferences (e.g., "Would the user prefer item A over item B given their history?") or solicit generative summaries of latent preferences (e.g., "Describe the user's taste profile based on past interactions.").

The process might involve leveraging an LLM to generate a broad suite of potential questions tailored to the specific domain and task, from which a human designer can then curate, refine, or further specialize the set, significantly augmenting the human designer's capacity to craft a comprehensive and effective question set.

By injecting such structured and high-level priors into the question design phase, the DHR learning process is more directly guided towards capturing the most salient information for effective decision-making and interpretability, aligning the representation with the core objectives of the task.

### B.3    COLD-START SCENARIOS

In cold-start scenarios, where a new user has a sparse interaction history, the DHR framework remains robust. The DHR encoder $E$ is trained on population data, allowing it to learn generalizable patterns and priors regarding user preferences. When presented with a short history $h_{sparse}$, the encoder leverages these learned priors to generate a profile $z$. Just as a standard recommender model infers "likely future items" based on limited data, the DHR infers "likely answers to profile questions." The QA-space provides a structured format for encoding these priors, enabling the system to generate meaningful initial profiles even with minimal user-specific data.

### B.4 ADVERSARIAL TRAINING OF THE QA GENERATOR

While our main results use a fixed QA-space, we also conducted experiments where the QA generator itself was learned. To achieve this, we formulated the task as an adversarial learning problem, where a QA agent, or generator, learns to pose questions and provide answers that are maximally informative or challenging for the DHR encoder and answer agent.

The original optimization problem (Eq. (OPT 1)) is designed to maximize task reward while minimizing the divergence between the DHR's predicted answer distribution and a fixed, ground-truth answer distribution. To learn the QA generator, we introduced an adversarial objective.

We refine the objective in Eq. (OPT 1) as follows:

$$\max_{E, \nu_A, \pi_D} \min_{\nu_{QA}^*} (1 - \lambda)V(\pi) - \lambda D_f \left( d^{\nu_A^*} \middle\| d^{\nu_A} \right),$$

(OPT 3)

where the adversarial QA generator $\nu_{QA}^*$ (which generates $(q, y)$ pairs from $h, \omega$) tries to select questions and answers to maximize the divergence $D_f$, making it harder for the DHR $E$ and answer agent $\nu_A$ to match its outputs. The DHR encoder $E$ and answer agent $\nu_A$ continue to try to minimize this divergence while maximizing task reward.

The dual variational form, analogous to Eq. (OPT 2), becomes

$$\max_{\substack{\pi_D, E, \nu_A}} \min_{\substack{g: \mathcal{H} \times \mathcal{Q} \times \mathcal{Y} \mapsto \mathbb{R} \\ \nu_{QA}^*: \mathcal{H} \times \Omega \mapsto \Delta_{\mathcal{Q} \times \mathcal{Y}}}} \mathbb{E} \left[ (1 - \lambda)r(h, a) + \lambda \mathbb{E}_{\substack{q \sim \nu_Q^*(h, \omega) \\ y \sim \nu_A(q, h, \omega)}} \left[ f^*(g(h, q, y)) \right] - \lambda \mathbb{E}_{\substack{q \sim \nu_Q^*(h, \omega) \\ y \sim \nu_A^*(z, q)}} [g(h, q, y)] \right].$$

(OPT 4)

In this formulation, $\nu_{QA}^*$ is the adversarial QA generator that produces $(q, y)$ pairs given history $h$ and future $\omega$. The agent's answer network $\nu_A$ predicts $\hat{y}$ for a question $q$ (generated by $\nu_{QA}^*$) using the DHR $z = E(h)$. The discriminator $g$ tries to distinguish between answers from $\nu_A$ and $\nu_{QA}^*$. The adversarial QA generator $\nu_{QA}^*$ is trained to generate $(q, y)$ pairs that are hard for $\nu_A$ to predict correctly, effectively maximizing the objective for $g$. Eq. (OPT 4) maintains strong duality, allowing for an iterative training algorithm where we updated $\pi_D, E, \nu_A, g$, and $\nu_{QA}^*$.

Our experiments with this adversarial setup (results in Table 3) showed a marginal reward improvement of 2-3%. This demonstrates the potential benefits of learning the QA-space, such as the automated discovery of more informative questions. However, we also found that ensuring the learned questions remained semantically meaningful and interpretable was a significant challenge. Without strong regularization (e.g., encouraging question diversity or biasing towards human-understandable questions), the generator could find trivial or uninformative ways to maximize the divergence. Given the substantial effectiveness of the fixed QA-generator bootstrapped with a powerful LLM and the added complexity of adversarial training, we conclude that the fixed approach remains highly practical.

Learning an adversarial QA generator is a complex but promising research direction which may significantly enhance the adaptability of the DHR framework. Our current work lays the foundation by demonstrating the effectiveness of DHRs given a well-defined QA-space, and future work can build upon this to explore dynamic and learned QA-spaces more thoroughly.

Table 3: Comparison of DHRL with a fixed QA-generator versus an adversarially trained QA-generator for the Gemma V3 12B model.

| Dataset | Model & Method | Prediction Accuracy | Rec. Reward ($r^D$) |
|---|---|---|---|
| Amazon | Gemma V3 12B + DHRL (fixed QA) | 0.75 | 0.84 |
| | Gemma V3 12B + DHRL (adversarial QA) | **0.77** | **0.86** |
| MovieLens | Gemma V3 12B + DHRL (fixed QA) | 0.84 | 0.93 |
| | Gemma V3 12B + DHRL (adversarial QA) | **0.86** | **0.95** |

## C  IMPLEMENTATION DETAILS

This appendix provides additional details regarding the implementation of our DHR learning (DHRL) framework.

**Models and Parameterization**    The DHR encoder ($E$), answer agent ($\nu_A$), and decision agent ($\pi_D$) are implemented using LLMs. As mentioned in the main paper (Section 5.1), primary LLM models include Gemini 1.5 Flash and Gemma V3 (4B, 12B). The DHR encoder and answer agent share the same base LLM architecture. A fixed anchor LLM (typically the original or supervised fine-tuned version of the model), is used to provide the reference distribution for the KL-divergence regularization term in the policy loss. The value network, responsible for estimating advantages and providing outputs for the DHR discriminator ($g$), defaults to a Gemma V3 4B model unless specified otherwise. The discriminator $g$ outputs are derived from the value network. This is achieved by feeding the value network QA pair samples (one reference, one target, as required by DHR). The value network's output vocabulary is conceptually partitioned: one segment is used for standard value prediction, while another segment provides the scalar outputs leveraged by the discriminator logic.

**Training Framework and Algorithm**    The core learning algorithm is implemented as an actor-critic method. The DHR encoder acts as the policy, and the answer agent components (value function and discriminator logic) form parts of the critic and learning signal. Training is run for up to 3000 optimization steps. As noted in the main paper, convergence was generally observed within 1000 iterations, which forms the basis for our reported results.

The multi-agent training paradigm introduces computational overhead compared to standard supervised fine-tuning. In our experiments, the DHRL training procedure resulted in an approximately 3x increase in training time compared to the predictive baseline (Gemma V3 Baseline).

**Key Hyperparameters**    Several hyperparameters governed the training process. These are summarized in Table 4.

**Loss Function Details (DHR)**    The choice of $f$-divergence for the discriminator was crucial. The default was TV-distance, as reported in the main paper. KL divergence and Chi-squared divergence were also explored. The DHR encoder's policy was regularized using KL divergence w.r.t. the fixed anchor (SFT) model, with the KL weight ($\alpha$) annealing. The total loss driving updates included: a policy gradient loss term derived from advantages; a value function loss term; the KL regularization term for the policy; and the discriminator loss term. The advantages were computed based on a sum of the environment reward $r^D$ (related to downstream task performance) and the DHR answer reward $r^A$.

Table 4: Key Hyperparameters for DHRL Experiments.

| Hyperparameter | Default Value |
| --- | --- |
| DHR Balancing Factor ($\lambda$) | 0.01 |
| Batch Size | 128 |
| LLM Input Token Length | 6144 |
| User Profile ($z$) Max Token Length | 256 (default) |
| User History Length ($H$) | 10 (default) |
| Number of Ranking Questions ($K_q$) | 5 (default) |
| Policy Update Delay | 20 steps |
| Policy Warmup Steps | 20 steps |

## D ADDITIONAL RESULTS

Tables 6 and 7 provide the complete table of results from Table 1 with 95% confidence intervals. We also provide missing results for ablations for Amazon in Fig. 5, showing similar results to Fig. 3. We also show in Fig. 5 (left) the learning curves for the ablation over profile lengths. We found that the Amazon dataset in particular is very sensitive to longer profile lengths. Future work can further explore the effects of the DHR length and its relation to the choice of DHR templates with their predictive capabilities.

### D.1 ADDITIONAL ABLATION STUDIES

This section presents additional ablation studies investigating the sensitivity to the balancing hyperparameter $\lambda$, the necessity of joint optimization, and the sufficiency of the learned DHR.

**Sensitivity to $\lambda$.** The hyperparameter $\lambda$ in (OPT 1) balances the RL objective ($V(\pi)$) and the DHR learning objective (the $f$-divergence). We analyzed the sensitivity of the recommendation reward to different values of $\lambda \in \{0.01, 0.1, 0.5\}$ using the Gemma V3 12B model. As shown in Figure 4, the performance is robust across the tested range, with optimal performance achieved when $\lambda$ is between 0.01 (our default) and 0.1.

**Necessity of Joint Optimization (FT-QA).** To understand whether the performance gains are attributable to the structured information provided by the QA-generator or the joint optimization framework (OPT 1), we conducted an ablation study (FT-QA). In this ablation, we trained the model using only the QA objective ($\lambda = 1$), effectively performing supervised fine-tuning on the generated QA pairs. We then evaluated the resulting representation on the downstream recommendation task.

The results in Table 5 show that FT-QA significantly underperforms the full DHR method on both datasets. On MovieLens, FT-QA improves over the standard predictive baseline, suggesting the QA structure itself is beneficial. However, on Amazon, FT-QA performs worse than the baseline. Crucially, the joint optimization of DHR consistently outperforms both approaches by a significant margin. This confirms that balancing QA fidelity with reward maximization is essential for learning representations effective for decision-making.

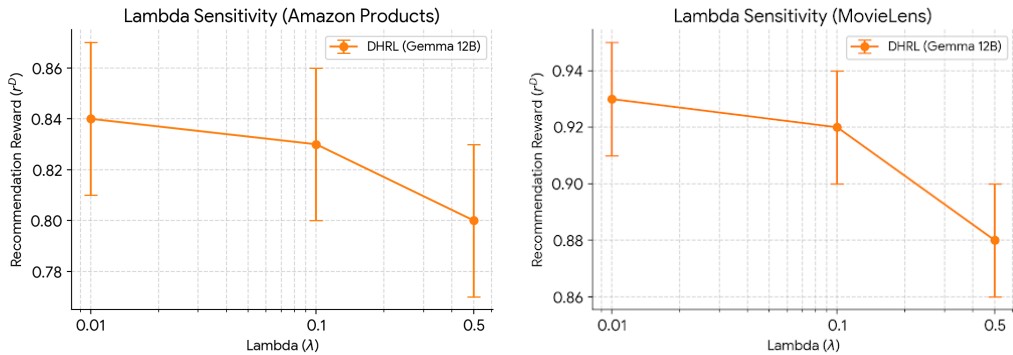

Figure 4: Sensitivity analysis of the recommendation reward to the balancing hyperparameter $\lambda$ on Amazon (left) and MovieLens (right) datasets.

Table 5: Ablation study on the necessity of joint optimization (Gemma V3 12B). FT-QA ($\lambda = 1$) uses only the QA objective. DHR uses the joint RL+QA objective.

| Dataset | Method | Prediction Accuracy | Rec. Reward ($r^D$) |
|---------|--------|---------------------|---------------------|
| Amazon | Baseline (Predictive) | 0.67 | 0.78 |
| | FT-QA ($\lambda = 1$) | 0.65 | 0.71 |
| | **DHR (Joint, $\lambda = 0.01$)** | **0.75** | **0.84** |
| MovieLens | Baseline (Predictive) | 0.58 | 0.64 |
| | FT-QA ($\lambda = 1$) | 0.72 | 0.78 |
| | **DHR (Joint, $\lambda = 0.01$)** | **0.84** | **0.93** |

**Testing DHR Sufficiency.**  To empirically test whether the learned DHR $z_t$ is a sufficient statistic of the history $h_t$, we conducted an ablation where the Decision Agent $\pi_D$ is given access to both the DHR and the raw history: $\pi_D(z_t, h_t)$. We compared this against the standard DHR approach $\pi_D(z_t)$. Using the Gemma V3 12B model, we observed a negligible improvement in recommendation reward (+0.5% on both datasets). This strongly suggests that the DHR $z_t$ captures nearly all the necessary information from $h_t$ for the decision-making task, closely approximating a sufficient statistic.

Table 6: Performance (with 95% CI) of DHRL with default settings (profile with max. 256 tokens, 5 questions, 10 history interactions, TV-divergence) after 1,000 iterations for Gemma V3 4B, 12B, and Gemini 1.5 Flash models.

| Model & Method | Prediction Accuracy (w.r.t. GT) | Rec. Reward ($r^D$) |
|---|---|---|
| Task: recommendation using Amazon products user profiles | | |
| Gemma V3 4B | $0.34 \pm 0.04$ | $0.54 \pm 0.04$ |
| Gemma V3 4B+DHRL | $0.71 \pm 0.04$ | $0.83 \pm 0.03$ |
| Gemma V3 12B | $0.67 \pm 0.04$ | $0.78 \pm 0.03$ |
| Gemma V3 12B+DHRL | $0.75 \pm 0.04$ | $0.84 \pm 0.03$ |
| Gemini 1.5 Flash | $0.69 \pm 0.04$ | $0.80 \pm 0.03$ |
| Gemini 1.5 Flash+DHRL | $0.74 \pm 0.04$ | $0.86 \pm 0.03$ |
| Task: recommendation using MovieLens user profiles | | |
| Gemma V3 4B | $0.37 \pm 0.04$ | $0.61 \pm 0.04$ |
| Gemma V3 4B+DHRL | $0.74 \pm 0.04$ | $0.79 \pm 0.03$ |
| Gemma V3 12B | $0.58 \pm 0.04$ | $0.64 \pm 0.04$ |
| Gemma V3 12B+DHRL | $0.84 \pm 0.03$ | $0.93 \pm 0.02$ |
| Gemini 1.5 Flash | $0.62 \pm 0.04$ | $0.68 \pm 0.04$ |
| Gemini 1.5 Flash+DHRL | $0.82 \pm 0.03$ | $0.92 \pm 0.02$ |

Table 7: Performance (with 95% CI) of DHRL with default settings (profile with max. 256 tokens, 5 questions, 10 history interactions, TV-divergence) after 1,000 iterations for Gemma V3 4B, 12B, and Gemini 1.5 Flash models.

| Model & Method | Profile-History Consistency (AI/Human) | Prediction Fidelity (AI/Human) | Review Quality (AI/Human) |
|---|---|---|---|
| Task: recommendation using Amazon products user profiles | | | |
| Gemma V3 4B | $3.46 \pm 0.16$ / $3.28 \pm 0.16$ | $0.44 \pm 0.04$ / $0.47 \pm 0.04$ | $0.15 \pm 0.03$ / $0.27 \pm 0.04$ |
| Gemma V3 4B+DHRL | $3.41 \pm 0.16$ / $3.85 \pm 0.15$ | $0.56 \pm 0.04$ / $0.53 \pm 0.04$ | $0.85 \pm 0.03$ / $0.73 \pm 0.04$ |
| Gemma V3 12B | $4.32 \pm 0.12$ / $4.18 \pm 0.13$ | $0.34 \pm 0.04$ / $0.39 \pm 0.04$ | $0.17 \pm 0.03$ / $0.35 \pm 0.04$ |
| Gemma V3 12B+DHRL | $4.39 \pm 0.12$ / $4.37 \pm 0.12$ | $0.66 \pm 0.04$ / $0.61 \pm 0.04$ | $0.83 \pm 0.03$ / $0.65 \pm 0.04$ |
| Gemini 1.5 Flash | $4.41 \pm 0.12$ / $4.30 \pm 0.13$ | $0.38 \pm 0.04$ / $0.41 \pm 0.04$ | $0.17 \pm 0.03$ / $0.38 \pm 0.04$ |
| Gemini 1.5 Flash+DHRL | $4.46 \pm 0.11$ / $4.34 \pm 0.12$ | $0.62 \pm 0.04$ / $0.59 \pm 0.04$ | $0.83 \pm 0.03$ / $0.62 \pm 0.04$ |
| Task: recommendation using MovieLens user profiles | | | |
| Gemma V3 4B | $3.72 \pm 0.15$ / $3.96 \pm 0.14$ | $0.32 \pm 0.04$ / $0.45 \pm 0.04$ | - |
| Gemma V3 4B+DHRL | $3.84 \pm 0.15$ / $4.24 \pm 0.13$ | $0.68 \pm 0.04$ / $0.55 \pm 0.04$ | - |
| Gemma V3 12B | $4.66 \pm 0.09$ / $4.58 \pm 0.10$ | $0.35 \pm 0.04$ / $0.42 \pm 0.04$ | - |
| Gemma V3 12B+DHRL | $4.62 \pm 0.10$ / $4.71 \pm 0.09$ | $0.76 \pm 0.04$ / $0.66 \pm 0.04$ | - |
| Gemini 1.5 Flash | $4.69 \pm 0.09$ / $4.53 \pm 0.11$ | $0.38 \pm 0.04$ / $0.39 \pm 0.04$ | - |
| Gemini 1.5 Flash+DHRL | $4.74 \pm 0.08$ / $4.70 \pm 0.09$ | $0.77 \pm 0.03$ / $0.62 \pm 0.04$ | - |

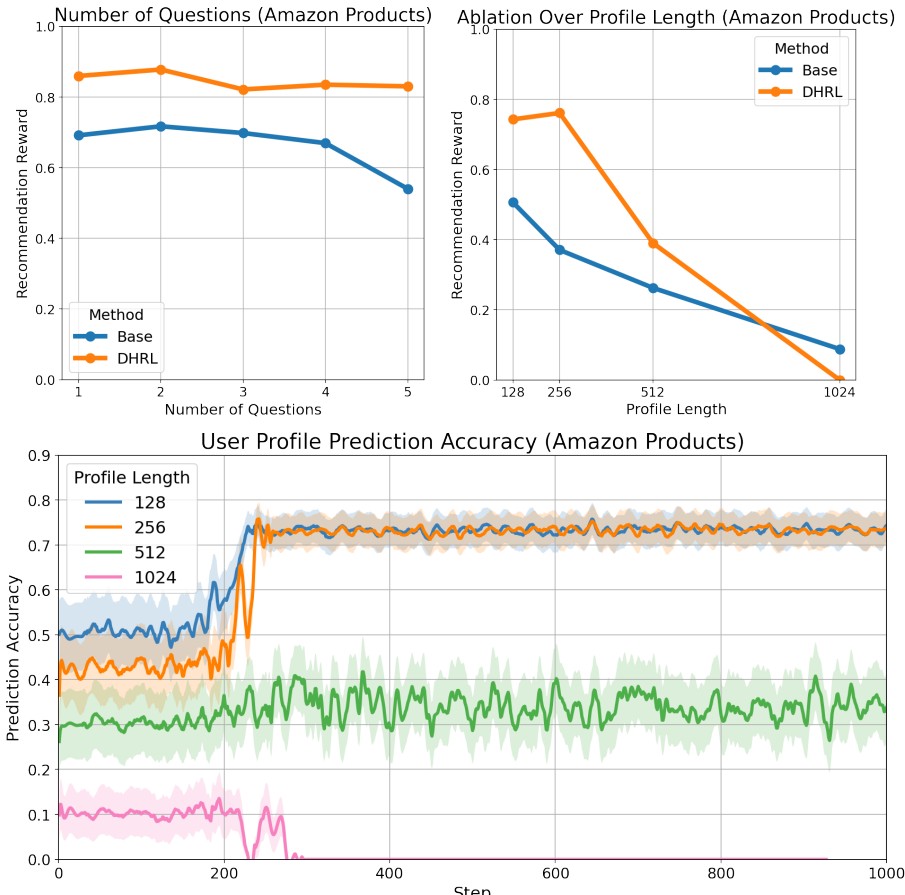

Figure 5: Ablation study on the history length (left) and profile length (right) the Amazon dataset, showing recommendation reward. Bottom plot shows learning curves for different profile lengths. Lower profile lengths were critical to ensure convergence.

# E  PROOF OF THM. 1

**Theorem 1.** *Let $E : \mathcal{H} \mapsto \mathcal{Z}$ be a DHR (Definition 3). Then it is also an $f$-sufficient statistic.*

*Proof.* Let $E_{DHR} : \mathcal{H} \to \mathcal{Z}$ be a Descriptive History Representation. We aim to show that $E_{DHR}$ is an $f$-sufficient statistic. According to the definition of $f$-sufficiency (Section 2), this means there exists a function $g^\pi : \mathcal{Z} \mapsto \mathbb{R}$ such that for any history $h \in \mathcal{H}$ and policy $\pi \in \Pi$:

$$f^\pi(h) = g^\pi(E_{DHR}(h)). \tag{1}$$

We are given an $f$-sufficient QA-space (Definition 2). This means for any history $h$, there is a set of sufficient questions $Q_h^*$. The set of sufficient question-answer pairs is defined as:

$$S_{QA}(h) := \{(q, \nu(h, q)) \mid q \in Q_h^*\}.$$

Since $S_{QA}(h)$ is defined as an $f$-sufficient statistic, there must exist a function $F^\pi$ such that:

$$f^\pi(h) = F^\pi(S_{QA}(h)). \tag{2}$$

We are also given that $E_{DHR}$ is a DHR (Definition 3). This implies the existence of an answer function $\nu_A$ such that for any $h$ and any $q \in Q_h^*$:

$$\nu_A(E_{DHR}(h), q) = \nu(h, q). \tag{3}$$

We now construct the required function $g^\pi(z)$. We must define $g^\pi(z)$ such that it depends only on $z$ (and $\pi$), not on a specific history $h$.

Let $z \in \mathcal{Z}$. We define the set of constructible QA pairs from $z$, denoted $S_{constr}(z)$. This set includes all QA pairs where the question $q$ is sufficient for *at least one* history $h'$ that maps to $z$, and the answer is generated by $\nu_A$ using $z$:

$$S_{constr}(z) := \{(q, \nu_A(z, q)) \mid \exists h' \in \mathcal{H} \text{ s.t. } z = E_{DHR}(h') \wedge q \in Q_{h'}^*\}.$$

We define $g^\pi(z)$ using the function $F^\pi$ from the sufficiency definition (Eq. 2):

$$g^\pi(z) := F^\pi(S_{constr}(z)). \tag{4}$$

To prove the theorem (Eq. 1), we must verify that $g^\pi(E_{DHR}(h)) = f^\pi(h)$. Let $z_h = E_{DHR}(h)$. Using Eq. 2 and Eq. 4, the proof reduces to showing:

$$F^\pi(S_{constr}(z_h)) = F^\pi(S_{QA}(h)).$$

We analyze the relationship between the two sets $S_{QA}(h)$ and $S_{constr}(z_h)$.

**Step A: Show $S_{QA}(h) \subseteq S_{constr}(z_h)$.**

Let $(q_0, y_0) \in S_{QA}(h)$. By definition, $q_0 \in Q_h^*$ and $y_0 = \nu(h, q_0)$. By the DHR property (Eq. 3), $y_0 = \nu_A(z_h, q_0)$.

To show $(q_0, y_0) \in S_{constr}(z_h)$, we must satisfy the condition in the definition of $S_{constr}(z_h)$: $\exists h'$ s.t. $z_h = E_{DHR}(h')$ and $q_0 \in Q_{h'}^*$. We can simply choose $h' = h$. Since $z_h = E_{DHR}(h)$ and $q_0 \in Q_h^*$, the condition is met. Thus, $S_{QA}(h) \subseteq S_{constr}(z_h)$.

**Step B: Establish the equality of $F^\pi$.**

It is possible that $S_{constr}(z_h)$ contains more QA pairs than $S_{QA}(h)$. (This occurs if another history $h'' \neq h$ maps to $z_h$, and $Q_{h''}^* \not\subseteq Q_h^*$).

However, $S_{QA}(h)$ is, by definition, a sufficient statistic for $f^\pi(h)$. A fundamental property of sufficient statistics is that augmenting a sufficient set of information with redundant but consistent information does not change the resulting value.

We must ensure $S_{constr}(z_h)$ is consistent. By its definition, all answers in $S_{constr}(z_h)$ are generated by the single function $\nu_A(z_h, \cdot)$. Therefore, for any question $q$, the answer is uniquely determined by $z_h$, ensuring consistency across the set.

Since $S_{QA}(h) \subseteq S_{constr}(z_h)$, $S_{constr}(z_h)$ is consistent, and $S_{QA}(h)$ is sufficient, it follows that:

$$F^\pi(S_{QA}(h)) = F^\pi(S_{constr}(z_h)).$$

We have shown that $f^\pi(h) = F^\pi(S_{QA}(h)) = F^\pi(S_{constr}(z_h)) = g^\pi(z_h)$. Therefore, $f^\pi(h) = g^\pi(E_{DHR}(h))$. This completes the proof that $E_{DHR}$ is an $f$-sufficient statistic. $\square$

# F LLM PROMPTS

Below we provide the complete prompts used in DHRL. The prompts are provided for the Amazon domain. The prompt used for Movielens is identical except for removal the review question.

Text in blue refers to parts of the prompt that are not changed. Text in red refers to placeholders that are replaced with e.g., titles, descriptions, user profiles, etc.

## F.1 DHR ENCODER

Your task is to write a user profile in the shopping domain for Clothing, Shoes, and Jewelry. The user profile should describe the user's preferences well.
Your user profile will later be used to predict the user's preferences and reviews over new items.
——————————————————————

You will be given a list of products the user has rated (ratings 1 to 5, where 5 is the highest rating).
You will also be given reviews for some of these products.
You will then be asked to write a user profile describing the preferences of the user.
——————————————————————

Here is an example of a user profile given a history:
#BEGIN EXAMPLE

User History:
Item #1
Item title: Robert Graham Men's Shipwreck Long Sleeve Button Down Shirt
Item description: N/A
Item price: N/A
User rating: 5.0
User review: Fantastic shirt. Great price. I saw the same shirt at Nordstrom for 229.00

——————————————————————

Item #2
Item title: Robert Graham Men's Whitehorse Long Sleeve Button Down Shirt
Item description: "The geometric pattern on this dress shirt creates an optical illusion to enhance your wardrobe. Crafted from Egyptian cotton for comfort.", "The geometric pattern on this dress shirt creates an optical illusion to enhance your wardrobe. Crafted from Egyptian cotton for comfort."
Item price: N/A
User rating: 5.0
User review: Thank you for such a great deal on a great shirt

——————————————————————

Item #3
Item title: Robert Graham Men's Shawn-Long Sleeve Woven Shirt
Item description: "When is a polo not just a polo? When it's made from exceptional mercerized pique cotton and boasting our signature space dyed tipping at the collar to take it from ordinary to extraordinary."
Item price: N/A
User rating: 5.0
User review: I have never had so many complements on a shirt.

Thank you for the great deal

————————————————————

Item #4
Item title: Robert Graham Men's Redzone Pixel Patterned Button-Front Shirt with Convertible Cuffs
Item description: N/A
Item price: N/A
User rating: 5.0
User review: love it

————————————————————

Item #5
Item title: Robert Graham Men's Shipwreck-Long Sleeve Button Down Shirt
Item description: "Every stylish guy needs a sophisticated striped shirt in his arsenal. Crafted from premium Egyptian cotton with space dyed stripes and a pop of paisley embroidery at the cuffs and inside placket for a little extra edge, this shirt has you covered at the office or a weekend away"
Item price: N/A
User rating: 5.0
User review: love it

————————————————————

Item #6
Item title: Robert Graham Men's Maitai-Long Sleeve Button Down Shirt
Item description: "Rain check. No way. Pack a mai tai and check's in for sunny days. Ombre squares of crisp bright white, ocean blues and poolside purples along with a dash of sunny blue sky over embroidery are the perfect cocktail for any getaway"
Item price: N/A
User rating: 5.0
User review: love it

————————————————————

Item #7
Item title: Lucky 21 Women's Plus Size Cuffed Sleeve One Button Boyfriend Blazer
Item description: "This boyfriend blazer is perfect for office and work wear, but also for a night out with friends! It features one button and cuffed sleeves."
Item Price: $8.00 - $34.99
User rating: 1.0
User review: sent back

————————————————————

Item #8
Item title: Robert Graham Men's Seattle Long Sleeve Button Down Shirt
Item description: "Hombre fade window pane print with satin stitch window pane edging and floral embroidery woven. Has digital printed neck band, back yoke and under collar."
Item price: N/A

User rating: 5.0
User review: Great shirt. Robert graham is the king

_________________________________________

Item #9
Item title: EleQueen 925 Sterling Silver Full Cubic Zirconia Bridal Flower Stud Earrings 15mm
Item description: N/A
Item Price: $16.99
User rating: 5.0
User review: The wife likes

_________________________________________

Item #10
Item title: Leveret Women's Robe Soft Micro Fleece Plush Shawl Collar Bathrobe Robe (Size Small-XXLarge)
Item description: N/A
Item Price: $39.99 - $49.99
User rating: 5.0
User review: Wife likes

User Profile:

#PROFILE#

1. Brand Loyalty:
   - Extremely High Loyalty to Robert Graham: The vast majority (7 out of 10) of rated items are from this brand. All Robert Graham items are rated 5/5. This is the single strongest signal in the data. The user explicitly calls Robert Graham "king".
   - Open to Other Brands, but with Caution: Purchased items from Lucky 21 (blazer, rated poorly), EleQueen (earrings, positive), and Leveret (robe, positive). This suggests a willingness to explore, but success outside of Robert Graham is mixed.

2. Product Category Preferences:
   - Men's Shirts (Dominant): Specifically, long-sleeve button-down shirts are the primary interest. Focus is on dress shirts and slightly more casual shirts (like a refined polo).
   - Women's Gifts/Items (Secondary): Purchases include a women's blazer, earrings, and a robe. Reviews indicate these are likely gifts ("Wife likes"). The low rating on the blazer suggests a different preference set for women's clothing, or simply a bad experience with that specific item.
   - No interest shown in shoes, and lower interest in jewelry.

3. Style Preferences (Men's Shirts):
   - Bold Patterns and Colors: Descriptions of favored items include "geometric pattern," "ombre squares," "pixel patterned," "striped," "floral embroidery." This user is not afraid of visually interesting shirts.
   - High-Quality Materials: Mentions of "Egyptian cotton," "mercerized pique cotton," "satin stitch," and "925 Sterling Silver" (for the earrings) indicate a preference for premium materials.

- Details and Accents: Appreciates features like "paisley embroidery," "space-dyed tipping," "convertible cuffs," and "digital printed neckband." These small details are important.
- Business Casual, elevated.

4. Style Preferences (Gifts/Women's Items):
   - Classic/Elegant: The earrings are described as "Bridal Flower Stud" with Cubic Zirconia, suggesting a classic, somewhat formal style. The robe is a "Soft Micro Fleece Plush Shawl Collar," indicating a preference for comfort and a traditional design.
   - Less information than men's.

5. Price Sensitivity:
   - Value-Oriented, but Willing to Pay for Quality: The user repeatedly mentions "great deal" and compares the price favorably to Nordstrom's. This suggests a search for good value, but not necessarily the absolute lowest price. They are willing to pay for a Robert Graham shirt, implying a higher price bracket acceptance for favored brands.
   - Wide Price Range: Has purchased items ranging from under $10 (potentially the blazer, depending on the specific price within the range) to likely over $100 (based on the Nordstrom comparison for Robert Graham shirts).

6. Review Style:
   - Concise and Positive (when satisfied): Uses short phrases like "love it," "Fantastic shirt," "Great shirt."
   - Value-Focused: Often mentions price and deals in positive reviews.
   - Direct (when dissatisfied): Simply states "sent back" for the negative review.
   - Expressive: show appreciation and gratefulness.

7. Purchase Channel: The purchase channel is not described.

8. Location/Region: No address, etc. is listed.

9. Purchase Frequency: Cannot be determined from this data alone. More history would be needed.

10. . Inferred Gender and Demographic Most likely Male, purchasing gifts for female.

#END#

#END EXAMPLE
———————————————————
———————————————————
The above is just an example, you do not have to follow this template exactly.
You can try different ways to describe the user's preferences, and you can also be more comprehensive in your description, if needed.
———————————————————
Now it's your turn.
Below is a list of items the user has rated:
1. Title: **<item 1 title>**
Description: **<item 1 description>**
Price: **<item 1 price>**
Review: **<item 1 review>**

2. Title: **<item 2 title>**
Description: **<item 2 description>**
Price: **<item 2 price>**

Review: **<item 2 review>**

3. Title: **<item 3 title>**
Description: **<item 3 description>**
Price: **<item 3 price>**
Review: **<item 3 review>**

4. Title: **<item 4 title>**
Description: **<item 4 description>**
Price: **<item 4 price>**
Review: **<item 4 review>**

5. Title: **<item 5 title>**
Description: **<item 5 description>**
Price: **<item 5 price>**
Review: **<item 5 review>**

6. Title: **<item 6 title>**
Description: **<item 6 description>**
Price: **<item 6 price>**
Review: **<item 6 review>**

7. Title: **<item 7 title>**
Description: **<item 7 description>**
Price: **<item 7 price>**
Review: **<item 7 review>**

8. Title: **<item 8 title>**
Description: **<item 8 description>**
Price: **<item 8 price>**
Review: **<item 8 review>**

9. Title: **<item 9 title>**
Description: **<item 9 description>**
Price: **<item 9 price>**
Review: **<item 9 review>**

10. Title: **<item 10 title>**
Description: **<item 10 description>**
Price: **<item 10 price>**
Review: **<item 10 review>**
————————————————————

————————————————————
————————————————————
Write a user profile to describe this user.
Ignore information that seems irrelevant or not informative.
Limit the profile to a maximum of 166 words.
Even though the length of the example profile above might be different, there is a strict limit of 166 words to
your output.

Output should be in the following format:
#PROFILE# <user_profile> #END#

## F.2    ANSWER AGENT

Below is a profile of a user's preferences:

**<user profile (from DHR encoder)>**

_______________________________________

Below are 6 items the user hasn't rated yet:

1. Title: **<item 1 title>**
Description: **<item 1 description>**
Price: **<item 1 price>**

2. Title: **<item 2 title>**
Description: **<item 2 description>**
Price: **<item 2 price>**

3. Title: **<item 3 title>**
Description: **<item 3 description>**
Price: **<item 3 price>**

4. Title: **<item 4 title>**
Description: **<item 4 description>**
Price: **<item 4 price>**

5. Title: **<item 5 title>**
Description: **<item 5 description>**
Price: **<item 5 price>**

6. Title: **<item 6 title>**
Description: **<item 6 description>**
Price: **<item 6 price>**
_______________________________________

_______________________________________

Below are a set of 6 questions about the items above:
(Q1) Rank the items **<id 5>** and **<id 6>** based on the user's preferences.
(Q2) Rank the items **<id 6>** and **<id 2>** based on the user's preferences.
(Q3) Rank the items **<id 2>** and **<id 1>** based on the user's preferences.
(Q4) Rank the items **<id 1>** and **<id 3>** based on the user's preferences.
(Q5) Rank the items **<id 3>** and **<id 4>** based on the user's preferences.
(Q6) Write a review for item **<id 1>** as the user would write it.
_______________________________________

Each of the 6 questions either asks you to rank two items based on the user's preferences, or to write a review for an item in the way the user might write it.

For ranking questions, a higher rank means the user would rate the item higher.
If there's a tie, pick the order randomly.
For review questions, you should write a raw review for the item as the user would write it.

Formatting your answer:
Your output should be in the following format. For each ranking question you should output a line with its prediction.
For a review question, you should output a review as the user would write it for that item.

For question k your output answer should be in the following format:
(Ak) #PREDICTION# [item_id, item_id] #END#
if it's a ranking quesiton, and
(Ak) #PREDICTION# user review #END#
if it's a review question.

For the ranking question, the list is in order of the user's preferences for those item ids in that question.
For the review question, don't prefix it with anything else (like "User review"). Just write the raw review the user would write between the #PREDICTION# and #END#.

For example, assume you are asked in question 1 to rank item ids 1 and 2, and you believe the user would rate item 1 higher than item 2, and in question 2 you're asked rank item ids 1, 3, where you the think the user would rate item 3 higher than item 1. And assume in question 3 you are asked to write a review for item id 1, and let's say the user liked that item.
Then your complete output should be:
(A1) #PREDICTION# [1, 2] #END#
(A2) #PREDICTION# [3, 1] #END#
(A3) #PREDICTION# This shirt is great! I loved the color and the material. #END#

Output should be exactly in the above format. Do not output anything else.
Remember to use #PREDICTION# and #END# for every answer.

# G  RATER STUDY

Below we show examples of rater forms including the contexts and questions:

## Sample Rater From for Prediction and Review Accuracy using Amazon Reviews

We have the following summary about preference of a user:

This user favors comfortable, stylish women's clothing, particularly sandals and dresses. They appreciate value and are drawn to visually appealing designs, as evidenced by the high rating on the silver chain. A preference for easy-to-wear items is clear, with ratings of 4.0 on the sandals and dress. They seem to like simple, classic styles and are happy to purchase items within a reasonable price range. The 5/5 rating on the flip flops indicates a practical focus on comfort and style.

1. Title: BIADANI Women Button Down Long Sleeve Basic Soft Knit Cardigan Sweater Item Item description: "Casual, Elegant Fitted Knit Cardigan Sweaters That is Versatile and Made Out of High Quality Material and Lusciously Soft."

Price: $10.01 - $25.80

_______________

2. Title: Moxeay Women Sexy Sleeveless Spaghetti Strap Mini Club Dress

Item Item description: "Material: Polyester (soft fabric make you comfortable to wear) Style: Spaghetti strap mini dress Features: U neck/V neck, Sleeveless, Spaghetti strap,Backless, High Waist, Back zipper Dress length: Mini length Occasion: Casual, Beach, Club, Prom, Banquet, Party Evening Attractive style and beautiful design, sexy summer sleeveless dress Ladies Dress ONLY, other accessories photographed not included. U neck style size: S( US 0) Bust:25.19"Waist:24.40"Length:23.62"M(US 2) Bust:26.77"Waist:25.98"Length:24.01"L(US 4) Bust:28.34"Waist:27.55"Length:24.40"XL(US 6) Bust:31.49"Waist:29.92"Length:24.80"Lace V neck style size: S:Bust 32.28",Waist 27.55", Length 30.31"M:Bust 33.85", Waist 29.13", Length 30.70"L:Bust 35.43", Waist 30.70", Length 31.10"XL:Bust 36.22", Waist 31.49", Length 31.49"Embroidery V neck style: S:Waist 29.92",Bust 34.64",Length 36.22" M:Waist 31.49",Bust 37.00",Length 36.61" L:Waist 33.07",Bust 39.37",Length 37.00" XL:Waist 33.85",Bust 41.73",Length 37.40" NOTE: 1.Size is Asian sizes,pls allow 1-2 inch size deviation due to manual measurement. 2.Colors may slightly different due to the lighting and monitor. Package Content: 1 x Women dress (Packed in Moxeay designed outpacking!)"

Price: $14.99 - $21.99

_______________

3. Title: Miusol Women's Casual Flare Floral Contrast Evening Party Mini Dress

Item description: unknown

Price: unknown

_______________

4. Title: Damask Embossed Metal Business Card Case

Item Item description: "This business card case features an embossed gold damask pattern on both sides. The case holds up to 10 business cards securely and snaps shut when not in use. Measures 3.75"x 2.5"x 0.25"."

Price: $3.95

_______________

5. Title: Honolulu Jewelry Company Sterling Silver 1mm Box Chain Necklace, 14-36

Item Item description: "Nickel free sterling silver 1mm box chain. Comes in different sizes with a spring clasp. Rhodium finished to prevent tarnishing. Made in Italy. Gift box included. From Honolulu Jewelry Company, Honolulu, Hawaii."

Price: $8.99

_______________

6. Title: Milumia Women's Button up Split Floral Print Flowy Party Maxi Dress

Item description: unknown

Price: $18.99 - $35.99

_______________

(Q1) Rank the items id 2 and id 3 based on the user's preferences.

(Q2) Rank the items id 3 and id 6 based on the user's preferences.

(Q3) Rank the items id 6 and id 5 based on the user's preferences.

(Q4) Rank the items id 5 and id 4 based on the user's preferences.

(Q5) Rank the items id 4 and id 1 based on the user's preferences.

(Q6) Pick a review for item id 1 as the user would write it:

REVIEW1: Love,love,love the color of this cardi. Will be buying more from this seller. I also love the button detail on the sleeve/cuff. Would highly recommend!

REVIEW2: This cardigan is so soft and comfy! It's a great basic piece that goes with everything. The fit is perfect, and it's really easy to throw on. I got it for a steal – it's definitely worth the price.

## Sample Rater Form for Profile Consistency using Amazon Reviews

Given a user with the following purchase history:

1. Title: Zumba Carpet Gliders for Shoes
Item Item description: "Dont get stuck on the carpet floor.Zumba Carpet Glidersallows you to step, shake, swivel and spin on carpet with ease and reduced risk of injury."
Price: unknown
User Rating: 1.0

2. Title: Sterling Silver Leverback Earrings Black Pear Teardrop Made with Swarovski Crystals
Item Item description: "Black faceted crystal teardrops hang from sterling silver rings and leverback earwires. Solid 925 sterling silver, teardrops are approx 5/8 x 3/8 inches. Made with Swarovski Crystals. Gift box or organza bag included, color or style may vary. See Joyful Creations store for matching necklace."
Price: $17.99
User Rating: 4.0

3. Title: Wiipu fashion vintage luxurious pink color crystal brand designer statement women necklace(B379)
Item Item description: "Material:alloy rhinestone ,crystal Size: necklace ribbon chain is 35cm ,pendant is 17cm*12cm; shipping from China, usually take about 7–15days arrival,if you not accetp please don't order, thanks!"
Price: unknown
User Rating: 4.0

4. Title: Kooljewelry Sterling Silver Bead and Diamond-Cut Ball Station Necklace (14, 16, 18, 20, 22, 24, 30, or 36 inch)
Item description: unknown
Price: $25.99
User Rating: 5.0

5. Title: Dearfoams Women's Sequin Flat Slipper
Item Item description: "Dearfoams is a slipper brand with great awareness. They understand exactly what customers are looking for– from pampering your feet in cozy softness to keeping you one step ahead of the game."
Price: unknown

User Rating: 4.0

6. Title: Women's Chiffon Beachwear Dress Swimwear Bikini Cover-up Made in The USA
Item Item description: "SHORE TRENDZ quality constructed cover-up Chiffon dresses are created with lovely detail. Buy with confidence as they are MADE IN THE USA. These sexy Chiffon dresses have finished edges and lovely pattern detail! We appreciate you visiting and welcome you to check out our store at SHORE TRENDZ for more great items!!!"
Price: $11.99
User Rating: 1.0

7. Title: Dearfoams Women's Lurex Sweater Knit Ballerina Slipper
Item Item description: "This slipper features a cable knit upper with silver lurex, yarn pom embellishment with silver lurex, and elasticized throat line for secure fit. Brushed terry lining and insole, 10mm high density poly foam insole, with durable, skid resistant, TPR outsole.", "Dearfoams is a slipper brand with great awareness. They understand exactly what customers are looking for– from pampering your feet in cozy softness to keeping you one step ahead of the game."
Price: unknown
User Rating: 3.0

8. Title: Nine West Women's Able Synthetic Platform Pump
Item Item description: "Nine West offers a quick edit of the runways – pinpointing the must have looks of the season, and translating what is fun, hip, and of the moment. It is trend-right footwear that you will reach for in your closet again and again. Nine West is sure to be your trusted resource for everyday chic style."
Price: $29.99 - $68.98
User Rating: 5.0

9. Title: Annie Shoes Women's Devine Dress Pump
Item description: unknown
Price: unknown
User Rating: 4.0

10. Title: TinkSky Wedding Tiara Rhinestones Crystal Bridal Headband Pageant Princess Crown
Item description: unknown
Price: $8.99
User Rating: 3.0

We want to use the following summary to capture user preference from the above purchase history:

1. **Brand Affinity:** Shows a strong preference for Nine West and Annie Shoes, evidenced by the 5-star ratings. A secondary interest in Sterling Silver jewelry.

2. **Style:** Appreciates fashionable slippers and pumps; likely enjoys comfortable yet stylish footwear.

3. **Price Range:** Primarily purchases items in the $10-$50 range.

4. **Negative Feedback:** The low rating of the beachwear dress and the tiara headband suggests a critical eye towards embellishments and potentially lower quality.

5. **Purchase type** Most likely female.

Does the above summary faithfully capture user preference from their purchase history?

1 - definite no
2
3
4
5 - definite yes

## Sample Rater Form for Prediction Accuracy using MovieLens

We have the following summary about preference of a user:

This user demonstrates a preference for comedies with a strong comedic impact, leaning towards the more witty and absurdist side. They enjoy action and adventure elements woven into their comedic entertainment, showing an appreciation for thrill and humor combined. While the user appreciates lightheartedness, they are not averse to a more serious tone - the presence of a thriller and a few dramas suggests a willingness to explore different genres, although they seem to lean towards more lighthearted plots. While the user appears to enjoy older comedies, they don't shy away from more contemporary fare, indicating a versatile taste across various time periods. They likely seek out movies with a clear comedic focus rather than those with a heavy dramatic weight. The user isn't afraid to give lower ratings to movies they don't enjoy, suggesting a discerning taste and a desire for quality comedic entertainment.

We will ask questions based on the following list of movies. Please do your own research (using IMDB) if you are not familiar with those movies.

1. Title: Hangover, The (2009)

________________________________

2. Title: Old Boy (2003)

________________________________

3. Title: Sympathy for Mr. Vengeance (Boksuneun naui geot) (2002)

________________________________

4. Title: Let the Right One In (Låt den rätte komma in) (2008)

________________________________

5. Title: Spanking the Monkey (1994)

________________________________

6. Title: Visitor Q (Bizita Q) (2001)

________________________________

(Q1) Rank the movies id 2 and id 6 based on the user's preferences.

(Q2) Rank the movies id 6 and id 5 based on the user's preferences.

(Q3) Rank the movies id 5 and id 1 based on the user's preferences.

(Q4) Rank the movies id 1 and id 3 based on the user's preferences.

(Q5) Rank the movies id 3 and id 4 based on the user's preferences.

## Sample Rater Form for Profile Consistency using MovieLens

Given a user with the following history:

1. Title: RoboCop (1987)

User Rating: 3.5

2. Title: Chasing Amy (1997)

User Rating: 3.5

3. Title: Grosse Pointe Blank (1997)

User Rating: 3.0

4. Title: Arachnophobia (1990)

User Rating: 3.0

5. Title: Mary Poppins (1964)

User Rating: 1.5

6. Title: Ice Age (2002)

User Rating: 2.0

7. Title: No Country for Old Men (2007)

User Rating: 4.0

8. Title: Wayne's World (1992)

User Rating: 4.5

9. Title: Bad Boys (1995)

User Rating: 2.5

10. Title: Planet of the Apes (1968)

User Rating: 3.5

We want to use the following summary to capture user preference from the above history:

This user enjoys a mix of genres, primarily leaning towards action, comedy, and sci-fi. They demonstrate a preference for films from the 80s and 90s and exhibit a decent tolerance for older movies. Their ratings suggest they favor films with a good balance of action, humor, and engaging plots. The user is not averse to darker or more somber films as long as the storytelling is strong (e.g., No Country for Old Men). Their rating of 'Mary Poppins', however, shows a potential dislike for saccharine, overly-sentimental films. The user shows definite interest in classic sci-fi but is less enthusiastic towards animation and may be reluctant to watch family-oriented content.

Does the above summary faithfully capture user preference from their history?

1 - definite no
2
3
4
5 - definite yes

# H  QUALITATIVE RESULTS

## H.1  AMAZON PROFILE EXAMPLE #1

**User History:**

1. **Title:** New Balance Women's WW665 Walking Shoe
   **Item description:** "New Balance is dedicated to helping athletes achieve their goals. It's been their mission for more than a century to focus on research and development. It's why they don't design products to fit an image. They design them to fit. New Balance is driven to make the finest shoes for the same reason athletes lace them up: to achieve the very best.", "Get fit and stay chic with the WW665 walking shoe from New Balance. Ample mesh allows refreshing breathability while the cushy sole delivers comfort stride after stride. A sporty look and soft color palette make this an ideal find for your fitness routine."
   **Price:** $36.75 - $62.39
   **User Rating:** 5.0

2. **Title:** Dickies Women's Relaxed Fit Straight Leg Cargo Pant Fade &Wrinkle Resistant
   **Item description:** "Inseam 32 inches. Fit tip: For accuracy, measure yourself in your undergarments. Give all measurements in inches. If your measurements are between sizes, order the larger size."
   **Price:** $28.99 - $86.53
   **User Rating:** 2.0

3. **Title:** Simulated Pink Pearl Rondelle Stretch Bracelet Silver Plated Description:
   **Item description:** unknown
   **Price:** $12.99
   **User Rating:** 5.0

4. **Title:** Eye Catching Women Leather Bracelet Silver Color Beads Cuff Jewelry with Magnetic Clasp 7.5(Black)
   **Item description:** "Another eye catching Urban Jewelry bracelet,silver beads color make a stunning splash along a luxe multi midnight black leather bracelet with magnetic closure.Provides a touch of nature combined style... Guaranteed to add a unique, trendy touch to almost all outfitsShipping &Delivery From The USAUrban Jewelry is located in New York City and Offer Worldwide ShippingEstimated Delivery Time: In the United States 2-4 Business Days. About Urban Jewelry We have a passion for fashion. Our goal is to create a jewelry haven where you will find great quality, affordable prices and trendy pieces. Urban Jewelry is an exclusive brand specializing in upscale stainlesssteel silver and leather accessories for women, men and teenagers. Urban Jewelry is locatedin New York City and ships worldwide. From the runway to your home. Urban Jewelry collection features thelatest styles, unique pieces, which will make you happy and your loved ones as well..Eye Catching Women Leather Bracelet Silver Color Beads Cuff Jewelry with Magnetic Stainless Steel Clasp 7.5(Black)"
   **Price:** $9.90
   **User Rating:** 4.0

5. **Title:** Vikoros Women Flowy Lace Overlay Adjustable Strap Crop Top Tank Bustier
   **Item description:** "Sell by Vikoros Store, pls refer the specification and picture details"
   **Price:** unknown
   **User Rating:** 2.0

6. **Title:** FRYE Women's Molly D Ring Short Boot
   **Item description:** "Be the ring leader in the Molly D Ring Short boots by Frye0ae. Hammered full grain leather upper. Two buckle accents for a vintage look. Side zipper closure for easy on and off. Soft leather lining. Cushioned leather footbed for all-day comfort. Durable leather and

rubber outsole for added traction. Imported. Measurements: Heel Height: 1 in Weight: 1 lb 1 oz Shaft: 5 1ᵉ0442 in Product measurements were taken using size 8, width B - Medium. Please note that measurements may vary by size.", "The Frye Company is the oldest continuously operated shoe company in the United States. Founded in 1863 by John A. Frye, a well-to-do shoemaker from England, and family-run until 1945, Frye products have a long and illustrious history. Frye boots were worn by soldiers on both sides of America's Civil War, soldiers in the Spanish-American war, and by Teddy Roosevelt and his Rough Riders. When home-steading drew adventurous New England families to the West during the mid and late 1800's many of the pioneers wore Frye Boots for the long journey. Today Frye remains true to its roots with its line of heritage boots, but continues to innovate as it introduces chic new handbags, pumps, and sandals to its collection."
**Price:** $179.00 - $361.41
**User Rating:** 4.0

7. **Title:** totes Women's Zelus Snow Boot
**Item description:** "If you are looking for a premium fashionable cold weather boot, look no further than the Totes Zelus. A waterproof rubber shell keeps your feet warm and dry in all types of weather. Top of the line Metro fleece lined and a thick collar for that snuggly fit. Plus a front speed lace system and colored trim gives you the best of the best- The Totes Zelus"
**Price:** unknown
**User Rating:** 5.0

8. **Title:** totes Eric Black
**Item description:** "- Item: Totes Eric. - Upper: Polyurethane. - Style: Mens winter boot featuring a polyurethane upper - rear pull tab and full length zippers for easy entry - faux leather overlays - faux fur lining with padded upper - Thermolite insole. - Sole: Durable rubber lug outsole for traction. - Heel: 1 inch. - Lining:Fully faux fur lined for warmth. - Origin: Imported. - Color: Black."
**Price:** unknown
**User Rating:** 5.0

9. **Title:** Lonestar Jewelry Lacrosse Stick &Necklace
**Item description:** "Exclusive Allan Robin Lacrosse Necklace with Lacrosse Stick Charm in Ball Chain Known to be the best gift for a Lacrosse player or enthusiast. Suprise your family with this perfect gift. Beautiful Lacrosse gift for girls and it comes in an elegant cotton-filled jewelry box so pretty easy to store. It's lead & nickel free so no irritation to sensitive skin. They can wear it at any occassion or casually while doing day-to-day activities. This piece of modern fashion jewelry is unique enough to show your loved ones passion to the world, without being too flashy. A beautiful Lacrosse Stick Charm which Measures 1 1/4ẍ 3/8Ňot Only Is This Lacrosse Jewelry A Great Gift From Parents or Grandparents, But Would Also Be A Wonderful Gesture From A Coach To Their Students! Order Yours Now And Dont Forget To Check Out The Hundreds Of Other Unique Sports Gifts That We Offer For All Of The Athletes In Your Life!"
**Price:** unknown
**User Rating:** 5.0

10. **Title:** M&c Women's | Boyfriend Style & Chronograph Look | FC0385
**Item description:** "The Marciano & Co. house brand M&c offers customers superior value in an increasingly competitive market. The M&c product lines draw inspiration from the most popular designs and are available at a fraction of designer prices. The Marciano & Co. mission remains consistent; to provide customers with products that are novel in design and affordable without compromising quality."
**Price:** $14.99
**User Rating:** 3.0

**User Profile:**  This user demonstrates varied shopping interests, with a clear preference for practical and comfortable footwear and accessories for themselves and potentially others. A recurring theme is a fondness for items emphasizing ease and warmth, evidenced by the 5-star ratings for snow boots (Totes Zelus & Eric), walking shoes (New Balance), and a stretch bracelet (Simulated Pink Pearl).

While appreciating affordability (as revealed by purchases from brands like totes and M&c), the user isn't averse to investing in quality when it aligns with their needs, indicated by the 4-star rating for Frye boots. Style leans towards sporty-casual and practical, with a tolerance for some embellishments (beaded bracelet, lacrosse necklace – likely a gift). Negative ratings suggest an unwillingness to purchase ill-fitting (Dickies pants) or poorly-performing (Vikorous top) items. An unclear gender preference outside of sale of women's apparel.

## H.2   AMAZON PROFILE EXAMPLE #2

**User History:**

1. **Title:** Hand Crocheted Beanie (02)-Rasta
   **Item description:** "This beautifully thick, hand crocheted cotton beanie will brighten up your cold days with its unbeatable style and insulating warmth. The clean, round crown shape is achieved by 1 panel of tightly crochet cotton, of repeating black, green, yellow and red stripes. The base of the crown is made of an open crochet pattern with the band at the opening crocheted with elastic to give it a lot of stretch for a perfect fit around any head. This hat is available in other stripe patterns as well as in solid colors to meet your particular taste. Made of 100% cotton. Hand crocheted item. Crown measures 6 deep, 7 wide. ONE SIZE fits most, from sizes 6 - 7 5/8. Available in an array of colors. Imported."
   **Price:** $4.99
   **User Rating:** 3.0

2. **Title:** Intimo Men's Tricot Travel Pajama Set - Big Man Sizes
   **Item description:**Every man should have a comfortable Nylon Travel Boxers. Wrinkle- Free."
   **Price:** unknown
   **User Rating:** 5.0

3. **Title:** Men Purple Mesh Pocket Shorts Inner Drawstring Avail Size S-5X Item description: unknown
   Price: unknown
   **User Rating:** 5.0

4. **Title:** Breda Men's 1627-Gold Mitchell Multi Time Zone Watch
   **Item description:** "A great-looking timepiece from Breda, this watch utilizes an excellent blend of materials and mechanics to create a functional accessory with a stylish flair. With the perfect mix of style and comfort, this watch will quickly become one of the most popular members of your watch collection.", "The Mitchel Collection", "This three time zone, large face men 2019s watch is available in three colors.", "Breda Watches", "Breda. Original style. Non-singular aesthetic.", "A creative collective with a shared appreciation for design that tells more than one story. We believe in the freedom of self-expression through style. Breda was born when we poured our collective imagination and expertise into designing watches inspired by a global lens. An eclectic unit of ambitious artists, designers, business-brains, photographers, innovators and style rebels, we've come together to create pieces that intrigue, inspire and challenge the expected.", "Our process of creation is our own unique alchemy. A key principle is that of discovery: we live contemporary culture, explore the past and dream up the future, scour the world's fashion stages and streets to challenge and inspire each other's imaginations. Then we design.", "We work with global partners to source the latest materials. With meticulous attention to detail, each innovative product is born with the purpose of transcending the ordinary."

**Price:** unknown
**User Rating:** 5.0

5. **Title:** Breda Men's 1627-silver Mitchell Multi Time Zone Watch
   **Item description:**", "A great-looking timepiece from Breda, this watch utilizes an excellent blend of materials and mechanics to create a functional accessory with a stylish flair. With the perfect mix of style and comfort, this watch will quickly become one of the most popular members of your watch collection.", "The Mitchel Collection", "This three time zone, large face menŽ2019s watch is available in three colors.", "Breda Watches", "Breda. Original style. Non-singular aesthetic.", "A creative collective with a shared appreciation for design that tells more than one story. We believe in the freedom of self-expression through style. Breda was born when we poured our collective imagination and expertise into designing watches inspired by a global lens. An eclectic unit of ambitious artists, designers, business-brains, photographers, innovators and style rebels, we've come together to create pieces that intrigue, inspire and challenge the expected.", "Our process of creation is our own unique alchemy. A key principle is that of discovery: we live contemporary culture, explore the past and dream up the future, scour the world's fashion stages and streets to challenge and inspire each other's imaginations. Then we design.", "We work with global partners to source the latest materials. With meticulous attention to detail, each innovative product is born with the purpose of transcending the ordinary."
   **Price:** unknown
   **User Rating:** 2.0

6. **Title:** Breda Men's 1627-black Mitchell Multi Time Zone Watch
   **Item description:**", "A great-looking timepiece from Breda, this watch utilizes an excellent blend of materials and mechanics to create a functional accessory with a stylish flair. With the perfect mix of style and comfort, this watch will quickly become one of the most popular members of your watch collection.", "The Mitchel Collection", "This three time zone, large face menŽ2019s watch is available in three colors.", "Breda Watches", "Breda. Original style. Non-singular aesthetic.", "A creative collective with a shared appreciation for design that tells more than one story. We believe in the freedom of self-expression through style. Breda was born when we poured our collective imagination and expertise into designing watches inspired by a global lens. An eclectic unit of ambitious artists, designers, business-brains, photographers, innovators and style rebels, we've come together to create pieces that intrigue, inspire and challenge the expected.", "Our process of creation is our own unique alchemy. A key principle is that of discovery: we live contemporary culture, explore the past and dream up the future, scour the world's fashion stages and streets to challenge and inspire each other's imaginations. Then we design.", "We work with global partners to source the latest materials. With meticulous attention to detail, each innovative product is born with the purpose of transcending the ordinary."
   **Price:** unknown
   **User Rating:** 3.0

7. **Title:** Camouflage Cap, Color: Orange Camo, Size: One Size Item description: unknown Price: $8.66
   **User Rating:** 5.0

8. **Title:** LRG Men's Big-Tall Core Collection Nine Camo T-Shirt
   **Item description:**The Core Collection Nine Camo Tee is constructed of Jersey featuring a contrasting pattern and custom camo print expertly rendered throughout with custom tag on neckline and hem."
   **Price:** unknown
   **User Rating:** 5.0

9. **Title:** Rocawear Men's Big-Tall Gradient Black Hearts Item description: unknown Price: unknown
   **User Rating:** 5.0

10. **Title:** Kangol Men's Tropic Galaxy Cap
    **Item description:**The galaxy is a kangol interpretation of a classic 8-quarter cap"
    **Price:** $37.40
    **User Rating:** 5.0

**User Profile:**   This user demonstrates a strong preference for men's apparel, particularly casual wear. They consistently rate items favorably (mostly 5-star) suggesting a generally positive shopping experience and valuing style. Key interests include:

- **Streetwear & Camouflage:** A recurring theme of camouflage patterns and brands like LRG and Rocawear indicates a liking for streetwear aesthetics.
- **Caps & Hats:** Several purchases of caps (Kangol, camouflage, orange camo) indicate a strong interest in headwear.
- **Timepieces:** The purchase of the Breda watches, despite a mixed rating experience reveals an interest in watches.
- **Comfortable Loungewear:** The purchase of pajama sets highlights a preference for comfort.
- **Price Sensitivity:** The moderate rating on the beanie suggests a mindful of price.

The user appears to be male, likely younger, and influenced by contemporary trends. They are open to experimenting with different styles while maintaining a casual, streetwear-inspired aesthetic.

### H.3    AMAZON PROFILE EXAMPLE #3

**User History:**

1. **Title:** Rubik's Cube - Mens Hip To Be Square Soft T-shirt Small Off-white
   **Item description:** "From Rubiks Cube comes this soft cotton short sleeve T-Shirt in off-white that features The Rubiks Cube and says Hip To Be Square. Great T-Shirt for anyone who loves The Rubiks Cube."
   **Price:** unknown
   **User Rating:** 3.0

2. **Title:** Intimo Women's Printed Microfleece Pajama Pant
   **Item description:** unknown
   **Price:** $18.00
   **User Rating:** 5.0

3. **Title:** Minecraft Boys' Adventure Youth Tee
   **Item description:** "The first time you find yourself staring out at the expansive world of blocks laid out before you, the possibilities are truly endless. What will you choose to do in this biome, generated just for you you can build soaring castles, dig sprawling underground complexes, or maybe you'll set out to see what is just over the horizon. Whichever way you choose to play, each adventure will be your very own."
   **Price:** $14.50 - $28.99
   **User Rating:** 4.0

4. **Title:** Hollywood Star Fashion Casual Basic Women's Semi-Crop Camisole Cami Tank Top with Adjustable Straps
   **Item description:** "This is a long length Tank top Versatile Basic Spaghetti Strap Satin Trim Stretch Camisole Tank Yoga Everyday Active Adventure Travel Fitted Scoop neckline, adjustable shoulder straps Satin Trim Fully stretchable Please note: this top dose NOT feature a built-in shelf bra Body

length in size medium: 28" 95% Cotton, 5% Spandex Imported. Satisfaction guaranteed Returns accepted. We ship worldwide"

**Price::** $5.00 - $19.99
**User Rating:** 5.0

5. **Title:** MANDI HOME Hot Sale Wedding Fashion 925 Silver Plated Jewelry Set Big Hand Chain Bracelet Necklace Ring Stud Earings Eardrop Water Drops
**Item description:** "1pcs for each!"
**Price:** $6.35
**User Rating:** 2.0

6. **Title:** Fruit of the Loom Girls' Cotton Spaghetti Strap Sport Bra
**Item description:** "Fruit of the Loom girls' spaghetti strap sport bras provide comfort and support for any activity. The cotton and lycra materials give them a soft feel and also make them machine-washable. Give the girl in your life a good start with Fruit of the Loom. Designed to give her support while giving her room to grow, tried and true Fruit of the Loom underclothes are the perfect choice for your growing girl"
**Price:** $7.53 - $12.68
**User Rating:** 5.0

7. **Title:** Fruit Of The Loom Women's Breathable Underwear Multipack (Assorted)
**Item description:** "Our breathable cotton collection is designed with a soft, cotton-mesh fabrication that not only supports the flow of air but wicks away moisture to help you stay cool and dry all day long."
**Price:** $4.50 - $45.00
**User Rating:** 5.0

8. **Title:** Froomer Womens Winter Outwear Light Down Coat Hooded Jackets
**Item description:** unknown
**Price:** unknown
**User Rating:** 5.0

9. **Title:** Rockland Luggage 17 Inch Rolling Backpack, Pink Dot, Medium
**Item description:** "Rolling backpack with double skate wheels. Measures 13 inch lx10 inch wx17 inch h. Features include 2 zippered pockets on the front of the bag with one having a built in organizer. Zippered main compartment opens to large storage opening, and two mesh side pockets. Fabric handle at the top and telescoping pull handle. Bag made of durable polyester with metal telescoping handle and heavy duty plastic stand on bottom to keep the bag standing.", "R01-PINKGIRAFFE shown.", "", "", "", ""
**Price:** $3.24
**User Rating:** 5.0

10. **Title:** Bags for Less Large Clear Vinyl Tote Bags Shoulder Handbag (Royal Blue)
**Item description:** unknown

**User Profile:** This user demonstrates a preference for affordable, practical, and comfortable clothing and accessories, primarily for women and children. While they appreciate value, ratings suggest a focus on functionality over high-end brands, as evidenced by consistently positive reviews for items like Fruit of the Loom underwear and Fruit of the Loom sport bras. The 5-star rating for a Rockland luggage rolling backpack indicates an appreciation for travel-friendly and durable goods.

They show an interest in basic apparel and accessories (camisoles, pajamas). There's a slight openness to novelty items (Rubik's Cube shirt, Minecraft tee), but overall, their choices lean towards everyday essentials.

The lower rating on the jewelry set suggests a less enthusiastic response to more elaborate items. The user appears to gravitate toward comfort, utility, and reasonable pricing.

## H.4  AMAZON PROFILE EXAMPLE #4

**User History:**

1. **Title:** Naturalizer Women's Ringo Sandal
   **Item description:** "The Ringo is a slingback sandal that features an N5 Comfort System and a manmade outsole.", "Naturalizer was one of the first shoe brands that women could turn to for the feminine style they coveted and the comfort they thought was impossible to attain. Naturalizer's fresh, unpretentious designs are a smooth fit with your wardrobe, your life and your own unique style. Naturalizer promises style that makes you look good and feel good - always."
   **Price:** unknown
   **User Rating:** 4.0

2. **Title:** Aerosoles Women's Tapestry
   **Item description:** "Every girl needs a timeless pump like the Aerosoles Tapestry. Featuring a 3 covered heel, softly angled toe and classic lines for a tried-and-true look. Stunningly soft memory foam insole is stitched and cushioned for your comfort, while the flexible rubber sole with diamond pattern drinks up hard impact. Youll feel great all day long!", "Destined to be your new favorite, Tapestry from Aerosoles Women's offers professional polish for the office and beyond. Showcasing a classic silhouette, this pretty pump features a leather or fabric upper that slips on to reveal the unbelievable comfort from Aerosoles that you have come to know and love. A flexible rubber outsole and a modest heel add a tasteful touch to any ensemble."
   **Price:** $64.99
   **User Rating:** 2.0

3. **Title:** Eagle Creek Travel Gear Undercover Money Belt (Khaki)
   **Item description:** "Looking for money belts? Look no further than this simple waist-worn under-clothing solution. Keep important travel documents and personal identification items out of sight in this money belt. Its made of durable and lightweight rip-stop fabric with a moisture-wicking and breathable back panel. Complete with zippered pocket for secure organization and soft elastic waistband with strap keeper. When youre not wearing it, simply tuck the strap into the slip pocket on the back, which was conveniently created for waist strap storage. Travel solutions that make sense."
   **Price:** $15.85
   **User Rating:** 5.0

4. **Title:** Maidenform Women's Comfort Devotion Demi Bra
   **Item description:** "Magnificently smooth and supportive. Maidenform's Comfort Devotion Demi Bra features foam, contour underwire cups made of plush fabric and smoothing wings with super soft fabric on the inside.Line dry or lay flat to dry"
   **Price:** $15.80 - $94.21
   **User Rating:** 5.0

5. **Title:** Champion Women's Jersey Pant
   **Item description:** "Champion Jersey Pant with a rib waistband is just the right fit and look for everyday wear."
   **Price:** $13.20 - $72.06
   **User Rating:** 3.0

6. **Title:** Men's Cotton Casual Ankle Socks
   **Item description:** "", ""

**Price:** $15.90
**User Rating:** 5.0

7. **Title:** Champion Absolute Sports Bra With SmoothTec Band
**Item description:** "The Absolute workout bra solids and prints at a great value. This bra has a patented smooth tec band for the ultimate in chafe resistance and comfort. A must have for any gym bag."
**Price:** $7.93 - $48.00
**User Rating:** 5.0

8. **Title:** uxcell Men Point Collar Button Down Long Sleeves Plaid Detail Slim Fit Shirts
**Item description:** "Description:One mock pocket point collar button down long sleeves plaid detail slim fit shirt. Feature mock pocket, plaids detail for build up your special character. Buttoned point collar for standard button down shirt. Buttoned cuffs is fused to keep a crisp, dressy appearance. Soft touch, comfortable fabric which is comfort to wear in all season. Suitable for date, daily work, travel and everyday wear. Match with formal trousers or stylish denim pants to build up fashion casual look. Please check your measurements to make sure the item fits before ordering. Body Size Chart (in inches)International
**Price:** $12.24 - $19.81
**User Rating:** 2.0

9. **Title:** Mens Colorful Dress Socks Argyle - HSELL Men Multicolored Argyle Pattern Fashionable Fun Crew Socks Item description: unknown
**Price:** $11.99
**User Rating:** 5.0

10. **Title:** Marvel Comics Crew Socks Mens Mix Hulk Spiderman, The Avengers Large With Bonus One XWi Wrist Sweatband
**Item description:** "A simply Marvel-ous way to get your man to change socks regularly, each set of socks feature classic artwork from the comics themselves, with full color, detailed woven images of the likes of Avengers past and present, including The Hulk, Spider-Man, Iron Man and Captain America Designs: Assorted Contents: 98% polyester, 2% elastane Our guarantee: 100% official merchandise Care instructions: Machine washable Packaging: Hanger Brand: Marvel Comics Size: USA(7-12) UK(6-11) Includes One Xportsworld Innovations Wrist Sweatband (Color chosen randomly)Composition: 80% Polyester, 20% Cotton Size: Width 8cm x Length 9cm ( W 3.14In x L 3.54In)"
**Price:** unknown
**User Rating:** 4.0

**User Profile:** This user demonstrates a practical and value-focused shopping style with varied interests. They show a clear preference for comfortable and practical items, consistently rating items highly that prioritize these aspects. A strong inclination towards athletic wear and socks (Champion, Marvel socks) is evident, suggesting an active lifestyle or interest in sports-related apparel. They appreciate quality, as shown by the 4-star rating for Naturalizer sandals.

While primarily focused on apparel, the purchase of a money belt suggests travel interests. The mixed reviews (2-star for Aerosoles, 2-star for uxcell shirt) indicate potential sensitivity to fit or style outside their comfort zone. Maidenform bra and women's socks suggest occasional purchases of items for a female partner. The user appears to find good deals and prioritizes functionality over high-fashion trends.

H.5 MOVIELENS EXAMPLE #1

**User History:**

1. **Title:** Star Wars: Episode IV - A New Hope (1977)
   **User Rating:** 5.0
   ________________________________

2. **Title:** Star Wars: Episode V - The Empire Strikes Back (1980)
   **User Rating:** 5.0
   ________________________________

3. **Title:** Raiders of the Lost Ark (Indiana Jones and the Raiders of the Lost Ark) (1981)
   **User Rating:** 4.5
   ________________________________

4. **Title:** Matrix, The (1999)
   **User Rating:** 5.0
   ________________________________

5. **Title:** Aliens (1986)
   **User Rating:** 3.0
   ________________________________

6. **Title:** Usual Suspects, The (1995)
   **User Rating:** 4.0
   ________________________________

7. **Title:** Léon: The Professional (a.k.a. The Professional) (Léon) (1994)
   **User Rating:** 4.0
   ________________________________

8. **Title:** Star Wars: Episode VI - Return of the Jedi (1983)
   **User Rating:** 5.0
   ________________________________

9. **Title:** Dark Knight, The (2008)
   **User Rating:** 3.5
   ________________________________

10. **Title:** Godfather, The (1972)
    **User Rating:** 4.0

**User Profile:** This user enjoys intellectually stimulating and visually impressive films, favoring narratives with depth and complex characters. They appreciate cinematic excellence and are willing to engage with challenging themes, demonstrating a sophisticated taste. A strong preference exists for science fiction and action, particularly those with philosophical underpinnings, as evidenced by high ratings of "Star Wars" films and "The Matrix".

They appreciate well-crafted thrillers and crime dramas, like "The Usual Suspects" and "Léon: The Professional", valuing intricate plots and suspenseful storytelling. Classic films, notably "The Godfather", are also enjoyed, suggesting an appreciation for cinematic history.

The user isn't limited to specific genres, embracing a diverse range of eras and cultural influences, with a noticeable fondness for European cinema. They are comfortable with slower pacing and unconventional structures, valuing atmosphere and character development. Overall, this individual seeks movies that provoke thought and deliver lasting impressions.

H.6 MOVIELENS EXAMPLE #2

**User History:**

1. **Title:** Last of the Mohicans, The (1992)
   **User Rating:** 3.0
   —————————————————————

2. **Title:** Ghostbusters (a.k.a. Ghost Busters) (1984)
   **User Rating:** 4.0
   —————————————————————

3. **Title:** Star Wars: Episode I - The Phantom Menace (1999)
   **User Rating:** 5.0
   —————————————————————

4. **Title:** Police Academy 3: Back in Training (1986)
   **User Rating:** 3.0
   —————————————————————

5. **Title:** E.T. the Extra-Terrestrial (1982)
   **User Rating:** 3.0
   —————————————————————

6. **Title:** 13th Warrior, The (1999)
   **User Rating:** 4.0
   —————————————————————

7. **Title:** Being John Malkovich (1999)
   **User Rating:** 4.0
   —————————————————————

8. **Title:** Blair Witch Project, The (1999)
   **User Rating:** 3.0
   —————————————————————

9. **Title:** Bug's Life, A (1998)
   **User Rating:** 4.0
   —————————————————————

10. **Title:** Clerks (1994)
    **User Rating:** 5.0

**User Profile:** This user demonstrates a preference for films that are both engaging and thought-provoking. They appreciate a balance between entertaining narratives and deeper themes, often favoring stories with complex characters and layered plots. A strong interest in both action and comedy is evident, alongside a clear appreciation for science fiction and suspenseful thrillers.

The user enjoys films with strong directorial vision and is not afraid to explore unconventional storytelling techniques. They value realistic characters and are drawn to films that delve into the human condition, often with a focus on relationships and identity.

Despite a love for contemporary cinema, they also appreciate classic films, signaling a broad appreciation for cinematic history and diverse cultural influences. They are open to a range of emotional tones, from lighthearted to dramatic, and demonstrate a tolerance for challenging subject matter. Ultimately, this user seeks out movies that are memorable, intellectually stimulating, and leave a lasting impression.

### H.7 MOVIELENS EXAMPLE #3

**User History:**

1. **Title:** Batman (1989)
   **User Rating:** 4.0
   ___________________________________

2. **Title:** Apollo 13 (1995)
   **User Rating:** 3.0
   ___________________________________

3. **Title:** Pulp Fiction (1994)
   **User Rating:** 5.0
   ___________________________________

4. **Title:** True Lies (1994)
   **User Rating:** 5.0
   ___________________________________

5. **Title:** Die Hard: With a Vengeance (1995)
   **User Rating:** 5.0
   ___________________________________

6. **Title:** Aladdin (1992)
   **User Rating:** 3.0
   ___________________________________

7. **Title:** Ace Ventura: Pet Detective (1994)
   **User Rating:** 4.0
   ___________________________________

8. **Title:** Batman Forever (1995)
   **User Rating:** 5.0
   ___________________________________

9. **Title:** Shawshank Redemption, The (1994)
   **User Rating:** 3.0
   ___________________________________

10. **Title:** Fugitive, The (1993)
    **User Rating:** 4.0

**User Profile:**    This user enjoys action-packed, engaging films with a strong sense of adventure and spectacle. They have a pronounced taste for thrillers and superhero movies, evident in their high ratings for iconic films like **Batman** and **Die Hard**. A preference for visually impressive and exciting narratives is clear.

They appreciate well-constructed plots and fast-paced storytelling, enjoying films that deliver immediate entertainment. However, the user also values character-driven narratives, demonstrated by the consistent 4.0 rating for **The Shawshank Redemption**.

There's a clear appreciation for entertainment that is accessible and exciting. They generally gravitate towards movies with a positive tone and fun energy, similar to **Aladdin** and **Ace Ventura**. The user isn't overly concerned with complex themes, valuing thrills and excitement over deeper philosophical explorations. A preference for blockbuster entertainment is evident, alongside a fondness for the superhero genre.

## H.8    MOVIELENS EXAMPLE #4

**User History:**

1. **Title:** Godfather: Part III, The (1990)
   **User Rating:** 3.0
   _______________________________________

2. **Title:** Stepmom (1998)
   **User Rating:** 3.0
   _______________________________________

3. **Title:** Blair Witch Project, The (1999)
   **User Rating:** 3.0
   _______________________________________

4. **Title:** Haunting, The (1963)
   **User Rating:** 5.0
   _______________________________________

5. **Title:** Abbott and Costello Meet Frankenstein (1948)
   **User Rating:** 3.0
   _______________________________________

6. **Title:** Black Sabbath (Tre volti della paura, I) (1963)
   **User Rating:** 5.0
   _______________________________________

7. **Title:** General, The (1926)
   **User Rating:** 5.0
   _______________________________________

8. **Title:** Wings (1927)
   **User Rating:** 4.0
   _______________________________________

9. **Title:** Jail Bait (1954)
   **User Rating:** 3.0
   _______________________________________

10. **Title:** M (1931)
    **User Rating:** 5.0

**User Profile:**   This user enjoys films with a strong narrative and appreciates classic cinema, particularly those from the mid-20th century. They favor dramas and thrillers, often with complex characters and thought-provoking themes, demonstrating a sophisticated taste. A significant portion of their ratings fall within the $3.0 - 5.0$ range, indicating an ability to appreciate both well-executed and critically acclaimed movies.

They are open to diverse genres and cultural origins, showing a preference for European and international films. While they enjoy suspenseful and eerie stories like "The Haunting," they also appreciate lighter, comedic fare like "Abbott and Costello Meet Frankenstein." The user values strong directorial vision and cinematic artistry, favoring films that build atmosphere and explore deeper human experiences.

A tolerance for slower pacing and unconventional storytelling is evident. They're not afraid to give a lower rating to films that don't meet their expectations. The user consistently seeks out films that are visually engaging and emotionally resonant, demonstrating a deep appreciation for the art of filmmaking.