# OpenReview forum: "Descriptive History Representations: Learning Representations by Answering Questions"
_ICLR.cc/2026/Conference — Submitted to ICLR 2026_

### Official Review · Reviewer_q2La · 2025-10-29

**Soundness:** 2
**Presentation:** 3
**Contribution:** 2
**Rating:** 4
**Confidence:** 4

**Summary:**

This paper proposes Descriptive History Representations (DHRs), where an encoder maps interaction histories to compact representations characterized by their ability to answer task-relevant questions. Training couples a decision objective with an f-divergence aligning a learned answerer to a generator built from full trajectories including future outcomes. Experiments on MovieLens-25M and Amazon Reviews generate textual user profiles and report improvements in pairwise prediction accuracy and recommendation reward over LLM baselines and two RecSys models.

**Strengths:**

Clear modular framework with encoder, answerer, decision agent, and discriminator, coupled via a precise dual objective

Strong interpretability: Generates human-readable textual user profiles that can be inspected

Consistent empirical gains over prompt-only LLM baselines and specialized RecSys models across multiple LLM sizes

Useful ablations

Detailed appendices with complete prompts, rater instruments, implementation details, and qualitative examples help with reproducibility.

**Weaknesses:**

1. The paper's biggest problem is that the QA-generator sees future user interactions during training but obviously can't at test time. This means baselines only get user history while your method trains with future supervision. That does not seem right. Unless I misunderstood the setup. More concerning, the paper never explains how questions actually get generated at deployment when one does not have that future information. I'd also expect to see a baseline where you simply fine-tune an LLM on the same history-future pairs used to build questions, which would tell us whether the QA framework actually helps or if you're just benefiting from using more data.

The method fundamentally depends on ν*_QA(h,ω), which requires:

Access to future trajectories ω (only feasible offline/in-simulation)
Hand-designed question templates (Sec 5.1: "Rank the following movies...")
Domain-specific engineering for each new task

This creates a circular dependency: to learn sufficient representations, you need sufficient questions; to design sufficient questions, you need to know what's decision-relevant, but that is part of the learning objective. The paper acknowledges this is "challenging" (p.4) but provides no principled solution or failure mode analysis.

2. The recommendation reward metric assumes one can always check the user's true rating for whatever you recommend. That's not how it works. One probably needs inverse propensity scoring, or something similar. Without these, I can't tell if your method is actually better or just recommending popular items. There's no analysis of diversity or coverage either.

3.Where's the LLM that's fine-tuned on exactly the same supervision you use? That's the most obvious comparison.

4. The definitions are confusing. The paper notes that a set of question-answer pairs is a sufficient statistic. I don't quite follow this. The definition says the representation should answer questions in a set that depends on the history you're trying to encode. That seems like circular reasoning.

5. Looking at Tables 1-2, those confidence intervals could be quite close, there's no significance testing.

**Questions:**

How does one actually generate questions at test time when you don't have future outcomes?

Are you restricting actions to items with logged ratings? If so, why not use standard ranking metrics?

What stops the same items from appearing in both your QA training supervision and your evaluation set?

Please add an LLM baseline that's trained on the exact same supervision you use

The definitions seem circular. How do you determine which questions are sufficient when that depends on the representation you're learning?

Have you tested whether just adding the raw history to your representation improves things? That would tell us if it's actually sufficient.

---

> ### Author Response · Authors · 2025-11-15
>
> We sincerely thank you for your detailed critique, which helped us clarify crucial aspects of the experimental design and motivated valuable new ablations.
>
> ### W1, W3, Q1, Q4. Use of Future Information, Unfair Supervision, and Missing Baselines
>
> We apologize for the confusion regarding the information flow and have significantly revised Section 4 to clarify the setup. Please see our response below.
>
> 1. Information Access (Training vs. Inference): Our framework adheres to standard practices in sequential modeling and RL, often formalized as Learning Using Privileged Information (LUPI) [Vapnik & Vashist, 2009].
> Training (Offline): We use future information ($\omega_t$) to compute supervision signals. This is standard; e.g., in RL, the return $G_t$ (the future) is the target for $V(h_t)$.
> Inference (Test Time): The agent only uses the history $h_t$. The Decision Agent $\pi_D$ receives only $z_t=E(h_t)$.
> (Q1) How are questions generated at test time? They are not. The QA-Generator and Answer Agent are used only for training.
> 2. Fairness and the "Missing" Baseline (W3/Q4): Crucially, our baselines are trained using the **exact same raw information** ($h_t$ and $\omega_t$). The "Gemma V3 (Baseline)" is the requested baseline: the same LLM backbone, trained on the same trajectories, using a standard predictive objective (e.g., next-item prediction).
> 3. The DHR Difference (Structure, not Data): The difference is not what information is used, but how the supervision is structured. Baselines use raw predictive supervision. DHR structures this supervision into high-level, descriptive questions.
> 4. New Ablation: Decoupling QA and RL (FT-QA). You asked whether the benefit comes from the QA structure itself (informative targets from the LLM teacher) or the joint optimization. We trained a model using only the QA objective ($\lambda=1$), effectively performing supervised fine-tuning on the generated QA pairs (FT-QA). We compare this to the Baseline (standard predictive objective) and the full DHR (joint objective) using the Gemma V3 12B model (Rec. Reward):
>
> | Method | Amazon | MovieLens |
> | :--- | :---: | :---: |
> | Baseline (Predictive) | 0.78 | 0.64 |
> | FT-QA ($\lambda=1$) | 0.71 | 0.78 |
> | **DHR (Joint)** | **0.84** | **0.93** |
>
> On MovieLens, structuring supervision as QA (FT-QA) is superior to standard predictive training (Baseline). On Amazon, FT-QA underperforms the Baseline. However, in both cases, the DHR joint optimization significantly outperforms both FT-QA and the Baseline. This confirms that jointly optimizing the representation for both descriptive accuracy (QA) and task performance (RL) is crucial for learning effective representations (see the updated Appendix D).
>
> ### W1, W4, Q5. Circular Definitions and Practical Dependency.
>
> We have clarified Definition 2. The set of theoretically sufficient questions $Q_h$  is a property of the history $h$ and the environment dynamics; it exists independently of the representation $z$ we are learning. This breaks the definitional circularity. Practically, we use the QA-generator to approximate $Q^{*}_h$. We acknowledge that designing this approximation requires domain engineering (W1), but DHR provides a structured way to leverage powerful LLM priors for this task (see Appendix B.2).
>
> ###  W2. Recommendation Reward metric, IPS, and Diversity.
>
> We acknowledge the reviewer's point regarding the "Recommendation Reward" metric. This metric, based on the user's true rating, assumes full-information feedback. We chose this approach for two main reasons:
>
> - Our primary goal is to evaluate the quality and sufficiency of the learned representation (DHR). This standard offline metric [e.g., Steck, 2010] provides a direct assessment of whether the DHR accurately captures user preferences using the available ground truth.
> - This methodology is consistent with the evaluation protocols used by the baselines we compare against on these datasets.
>
> We agree that Inverse Propensity Scoring is necessary for unbiased evaluation in online (bandit) settings, and that diversity/coverage metrics are crucial for real-world deployment. We view these aspects of deployment evaluation as orthogonal to the focus of this paper (representation learning efficacy) and consider them important directions for future work.
>
> ###  W5. Statistical Significance.
> We have added 95% confidence intervals in Appendix D (Tables 6 and 7), confirming statistical significance (e.g., non-overlapping intervals for Amazon accuracy: $0.75 \pm 0.04$ DHR vs. $0.67 \pm 0.04$ Baseline).
>
> ### Q6. Testing Sufficiency
>
> Excellent suggestion. We conducted the requested ablation where the Decision Agent accesses both the DHR and the raw history: $\pi_D(z_t, h_t)$. We observed a negligible improvement (+0.5% reward) compared to $\pi_D(z_t)$. This strongly suggests that the DHR $z_t$ closely approximates a sufficient statistic. (Appendix D).

---

### Official Review · Reviewer_YkaA · 2025-10-30

**Soundness:** 2
**Presentation:** 3
**Contribution:** 3
**Rating:** 4
**Confidence:** 3

**Summary:**

Summary:

This paper introduces Descriptive History Representations (DHRs), a novel framework for learning informative representations of long interaction histories in partially observable environments. The core idea is to move away from traditional methods like belief states or predicting low-level observations. Instead, DHRs are learned by optimizing a representation's ability to answer a set of high-level, task-relevant questions about the past or future. The authors formalize this by defining a QA-space (Question-Answer-space) and posit that a representation that can successfully answer a set of "sufficient questions" acts as a sufficient statistic for the history. To learn these DHRs, a multi-agent cooperative framework called DHRL (Descriptive History Representation Learning) is proposed.


The key contributions of this paper are:
1. Formalization of DHRs: The paper introduces Descriptive History Representations (DHRs) as a new type of history representation defined by its ability to answer relevant questions. It formally links DHRs to the concept of f-sufficient statistics.

2. Multi-Agent Learning Framework (DHRL): It proposes a novel and flexible learning algorithm (DHRL) that operationalizes the DHR concept. This framework jointly trains an encoder, an answer agent, and a decision agent using a combined objective derived from a variational form of an f-divergence.

3. Interpretable Representations: The framework is shown to be capable of generating interpretable, textual user profiles that serve as effective representations for downstream tasks.

4. Empirical Validation: The paper provides empirical evidence on public recommendation datasets (MovieLens and Amazon) demonstrating that DHRL can outperform baseline methods in both prediction accuracy and recommendation reward.

**Strengths:**

* Novelty and Abstraction: The paper's main strength is its novel approach to representation learning. By focusing on answering high-level, semantically meaningful questions rather than predicting low-level observations, it shifts the representation burden to a more abstract and potentially more task-relevant level.

* Interpretability: In an age of black-box models, the ability to generate an interpretable textual user profile as the history representation is a significant advantage, particularly for domains like recommender systems.

* Flexible Framework: The DHRL algorithm is flexible, supporting different f-divergences (e.g., TV-distance, KL) and capable of operating in both online and offline training paradigms.

* Thorough Ablation Studies: The paper includes useful ablation studies analyzing the impact of history length, profile (DHR) length, and the number of questions, providing valuable insights into the model's behavior.

**Weaknesses:**

* Reliance on QA-Generator: The entire framework is critically dependent on the availability of a high-quality QA-generator ($\nu_{QA}^{*}$) to provide "sufficient" questions and ground-truth answers during training. The paper acknowledges that designing this oracle is challenging and relies on a pre-trained LLM for its main experiments. This dependency might limit its applicability in domains where such questions are hard to formulate or where a powerful pre-trained generator is unavailable.

* Training Complexity: The proposed DHRL framework is complex, involving the joint optimization of four components: the DHR encoder ($\theta_{E}$), the answer agent ($\theta_{A}$), the decision agent ($\theta_{D}$), and a discriminator ($g$). This could be computationally expensive and difficult to tune.

* Marginal Gain from Learned Generator: The paper reports that adversarially learning the QA-generator (as opposed to using a fixed one) "led to a marginal improvement in reward (2-3%)". This suggests that the framework for learning questions is not yet a major strength, and the success of the method hinges more on having good questions (e.g., from a pre-trained LLM).

* Limited Domain Exploration: The experiments are confined to recommender systems. While this is a suitable domain, the paper's claims about DHRs being a general approach for POMDPs would be much stronger if validated in other, more dissimilar domains (e.g., robotics, navigation, or game-playing).

**Questions:**

1. On the Objective Function: The paper proposes the joint objective in (OPT 1):$max_{E,\nu_{A},\pi_{D}}(1-\lambda)V(\pi)-\lambda D_{f}(d^{\nu_{A}^{*}}||d^{\nu_{A}})$. How was the hyperparameter $\lambda$ (which balances reward vs. question-answering) selected? How sensitive is the final performance of the decision agent $\pi_{D}$ to this trade-off?

2. On Attributing Performance: Since the main experiments use a QA-generator bootstrapped with a pre-trained LLM, how much of the performance gain over baselines is attributable to the novel DHRL framework versus the strong prior knowledge already embedded within the LLM used to generate questions and answers?

3. On Training Stability: Given the max-min optimization of the variational objective (OPT 2) involving a discriminator $g$ and three other agents, did you encounter any training instabilities? How does the choice of f-divergence ($D_{f}$) impact not only final performance (as shown) but also training stability?

---

> ### Author Response · Authors · 2025-11-15
>
> We appreciate the reviewer highlighting the novelty, abstraction, and interpretability of DHRs.
>
> ### Q2. Re Attributing Performance (DHRL vs. LLM prior)
>
> You asked if the gain is from the LLM priors in the QA generator (creating informative targets) or the joint optimization (OPT 1). To decouple these effects, we conducted a new ablation (**FT-QA**), training a model using only the QA objective ($\lambda=1$), effectively performing supervised fine-tuning on the QA targets. We compare this to the baseline (standard predictive objective) and the full DHR (Rec. Reward, Gemma 12B):
>
> | Method | Amazon | MovieLens |
> | :--- | :---: | :---: |
> | Baseline (Predictive) | 0.78 | 0.64 |
> | FT-QA ($\lambda=1$) | 0.71 | 0.78 |
> | **DHR (Joint)** | **0.84** | **0.93** |
>
> While the benefit of QA supervision alone (FT-QA vs. Baseline) varies by dataset, the full DHR (joint optimization) consistently and significantly outperforms both. This confirms that jointly optimizing the representation for both descriptive accuracy (QA) and task performance (RL) is crucial for learning effective representations (Appendix D).
>
> ### W2/Q3. Re Training Complexity & Stability.
>
> * **Complexity:** The complex training occurs offline; inference is efficient. We have quantified the training overhead: Training time increased approx. 3x over the baseline (See Appendix C).
> * **Stability (Q3):** We did not encounter instabilities (See Fig. 5 learning curves). Because the fixed QA-generator provides a stationary target distribution, the optimization (OPT 2) is a stable density ratio estimation task, rather than an unstable adversarial game where the target is moving (like GANs).
>
> ### W1/W3. Re QA-Generator
>
> We view the QA-generator as a mechanism to inject semantic priors. The fact that the adversarial generator yields marginal gains (Table 3) is a positive practical result: it demonstrates that off-the-shelf LLMs are sufficient to bootstrap an effective QA-space without complex adversarial training.
>
> ### Q1. Re Hyperparameter $\lambda$ selection and sensitivity
>
> We performed a grid search over $\{0.01, 0.1, 0.5\}$. We found the method to be robust, with values near $0.01-0.1$ performing optimally. We’ve added a sensitivity analysis plot to Appendix D.
>
> ### W4. Re Domain Exploration
>
> We agree this is an exciting future direction. DHRs generalize conceptually to other domains by framing questions as General Value Functions (GVFs) (we’ve added this connection to the *Related Work* section in our uploaded revision). Though we believe our empirical results in the recommendations domain are of significant importance, and demonstrate the effectiveness of our approach.

---

> ### Comment · Reviewer_YkaA · 2025-11-27
>
> Thanks for authors for replying my concerns. However, I still have one major question remaining:
> W1/W3: This dependency of an off-the-shelf pre-trained LLM might limit its applicability in domains where such questions are hard to formulate or where a powerful pre-trained generator is unavailable. And the failure in adversarial fine-tuning the QA generator for more effective QA space also suggests that this critical framework for learning questions and answers is not yet for widely customization, especially for domains where high quality QA pairs are lacking.
>
> Regarding this major concern not solved yet, I decide to keep my original rating (slightly negative).

---

> > ### Author Response · Authors · 2025-11-29
> >
> > We thank the reviewer for the continued engagement. We understand that due to the recent ICLR policy changes regarding the leak, scores cannot be adjusted further, but we wish to leave a final clarification regarding the concern that "dependency on an off-the-shelf LLM limits applicability."
> > ​
> >
> > We respectfully disagree with the premise that leveraging foundation model priors effectively is a limitation. On the contrary, we believe the empirical evidence provided in our rebuttal demonstrates that the DHR framework provides value specifically by how it leverages these priors, not just because they exist.
> >
> > ​1. The gain comes from the method, not just the LLM (Evidence: Table 5).
> > The reviewer suggests the success is due to the "strong prior knowledge already embedded within the LLM." We explicitly tested this in the FT-QA ablation (Appendix D.1).
> >
> > A. FT-QA ($\lambda$=1): We fine-tuned the same LLM on the same informative QA pairs using standard supervision.
> >
> > ​B. We trained using our proposed joint optimization (DHR joint)
> >
> > ​C. DHR significantly outperforms FT-QA (e.g., 0.84 vs 0.71 Rec. Reward on Amazon).
> > If the performance came solely from the "powerful pre-trained generator," FT-QA should have matched DHR. The performance gap proves that the joint optimization of the representation for decision-making is the source of our gain, effectively "unlocking" the LLM's potential where standard fine-tuning fails.
> >
> > ​2. The reviewer noted that adversarial training yielded marginal gains over the fixed generator. We view this as a positive practical result (strength), not a failure. It demonstrates that off-the-shelf LLMs are already sufficient to bootstrap high-performing RL representations without the instability and complexity of adversarial minimax training. This makes DHR more applicable to real-world industrial settings, not less.
> >
> > ​3. In domains without text (e.g., robotics), the "Question" space maps naturally to General Value Functions (GVFs) or auxiliary tasks (e.g., "Will I collide in 5 steps?"). The DHR framework remains valid; the source of the "questions" simply shifts from an LLM to domain-specific heuristics or GVFs.
> >
> > ​We hope this clarifies that DHR is a general methodology for structuring supervision, validated by the fact that it outperforms baselines given the exact same information and priors.

---

### Official Review · Reviewer_6zd7 · 2025-10-30

**Soundness:** 3
**Presentation:** 3
**Contribution:** 3
**Rating:** 6
**Confidence:** 2

**Summary:**

This paper introduces Descriptive History Representations (DHRs), a novel framework that learns to generate compact and interpretable history representations by answering semantically meaningful questions about user preferences and future behaviour. The authors propose a multi-agent learning algorithm that optimises a representation encoder, a decision policy and a question-answering module simultaneously.

**Strengths:**

1. The core idea is novel: defining representation sufficiency in terms of the ability to answer task-relevant questions offers a new, principled approach to interpretable representation learning.
2. The experimental evaluation is thorough, covering multiple datasets and metrics. The results show consistent improvements over standard LLM-based and specialised recommendation methods.
3. The practical framework leverages off-the-shelf LLMs as QA generators and supports offline training, enabling deployment in industrial settings.

**Weaknesses:**

1. The QA generator uses future user behaviour to construct questions and ground-truth answers. However, can the method work in online cold-start settings where new users have no explicit ratings or reviews? Without ground-truth answers to supervise the QA generator, it is unclear how meaningful questions could be constructed or how the DHR would learn.
2. The optimization objective is a complex min-max game involving the joint training of multiple components. Did the authors observe any training instability or mode collapse? How does the convergence speed of DHRL compare to a baseline that simply fine-tunes a single decision-making model?
3. Current experiments are concentrated on recommendation systems with rich textual interactions. How would the DHR framework be applied to partially observable environments where the observation space is non-linguistic? How can a meaningful QA-space be defined for these domains?

**Questions:**

The same as the weaknesses.

---

> ### Author Response · Authors · 2025-11-15
>
> We thank the reviewer for highlighting the novelty, thoroughness, and principled nature of our work. Please find our response to your comments and suggestion below.
>
> ### W2. Optimization instability or mode collapse.
>
> We did not observe instability or mode collapse (Fig. 5). We attribute this stability to the stationarity of our target distribution. In our primary setting (Fixed QA), the target $d^{\nu^*_{QA}}$ is fixed. While we use a min-max objective (OPT 2), the discriminator $g$ is solving a density ratio estimation task against a stationary target. This is significantly more stable than fully adversarial settings (like GANs/GAIL) where both distributions are moving.
>
> ### W1. Online cold-start settings.
> Our method is robust to cold-start scenarios. The DHR encoder is trained on population data, learning generalizable preference structures. For a new user with sparse history, the encoder leverages these learned priors. Just as a standard recommender infers "likely future items" from a short history, DHR infers "likely answers to profile questions" based on the same priors.
>
> We’ve added a detailed discussion on this in the new Appendix B.3. Furthermore, this conceptual argument is empirically supported by our analysis of history length (Figure 3, left), which shows that DHR maintains strong performance and significantly outperforms the baseline even with very short interaction histories (e.g., H=5).
>
> ### W3. Application to non-linguistic environments.
> In non-linguistic domains (e.g., robotics), the "questions" can be defined as specific state-predictive queries (e.g., "Is the robot stuck?"). The QA-space effectively becomes a set of General Value Functions (GVFs) or auxiliary tasks.

---

### Official Review · Reviewer_2xd4 · 2025-11-01

**Soundness:** 3
**Presentation:** 2
**Contribution:** 3
**Rating:** 6
**Confidence:** 4

**Summary:**

This paper introduces Descriptive History Representations, a framework for learning compressed representations of interaction histories in partially observable environments. It claims that an effective representation should be a sufficient statistic characterized by its capacity to answer questions about the past and future, rather than merely predicting low-level future observations. The authors propose a multi-agent learning algorithm that jointly optimizes a representation encoder, a decision agent, and an answer agent to maximize task reward while ensuring the representation can accurately answer questions.

Experimental results on public datasets show that the proposed method significantly outperforms base LLMs and state-of-the-art recommender systems in prediction accuracy and recommendation reward. The generated textual profiles are also rated highly by both AI and human evaluators, underscoring the method's strength in both performance and interpretability.

**Strengths:**

(1) The formulation of representation learning through question-answering is novel. It provides a interpretable alternative to methods that learn representations implicitly through prediction or reconstruction losses.

(2) The framework offers a high degree of interpretability. This is a significant advantage over existing methods.


(3) The paper provides comprehensive experiments on several datasets.  It includes extensive ablation studies (e.g., on history length, profile length, number of questions, choice of f-divergence) that provide valuable insights into the factors influencing performance.

**Weaknesses:**

(1) Dependence on a powerful QA-generator: The framework's performance is dependent on the quality and relevance of the questions generated by the oracle. While the use of a fixed LLM is practical, it introduces a dependency on the capabilities and potential biases of that specific model. The paper notes that adversarial training of the QA-generator yielded marginal gains, suggesting room for improvement in dynamically learning the optimal question set.

(2) Computational complexity: The multi-agent training paradigm, requiring the sampling of full trajectories and the training of multiple LLM-based agents, is computationally intensive.

(3) Limited evaluation scope: While the recommendation domain is a compelling and challenging testbed, the generalizability of the proposed method to other tasks and domains remains an open question and is not demonstrated.

(4) The proof of Theorem 1 relies on the assumption that the set of sufficient questions is a function of the representation itself. How to enforce it during learning is not deeply explored.

**Questions:**

Please find the problems in the Weaknesses part.

---

> ### Author Response · Authors · 2025-11-15
>
> We thank the reviewer for recognizing the novelty, interpretability, and comprehensive experiments of DHR. Please find our response to your comments below.
>
> ### W4. The proof of Theorem 1 assumption.
>
> Thank you for noticing this. We’ve revised the proof in Appendix E to remove this dependency entirely. The revised proof relies strictly on Definition 3 (the DHR property: $\nu_A(E(h), q) = \nu(h, q)$ for sufficient $q$) to show that the information captured by the sufficient QA pairs of $h$ is fully reconstructible from $E(h)$, thereby rigorously proving $E(h)$ is an f-sufficient statistic without the prior assumption.
> The key change involves demonstrating that even if different histories mapping to the same $z$ have different sets of sufficient questions, the DHR property (Definition 3) ensures that $z$ consistently captures the answers to all of them. We construct a consolidated set of QA pairs derivable from $z$. Since the original history's QA pairs are sufficient by definition, and they are contained within this consistent consolidated set captured by $z$, $z$ must also be sufficient. The revised proof shows that $E(h)$ is an f-sufficient statistic without the prior assumption.
>
> ### W2. Computational complexity.
>
> We clarify the distinction between training and inference costs. At inference, the system is efficient, requiring only the Encoder and Decision Agent. The multi-agent overhead (QA-Generator, Discriminator) is incurred only during (offline) training. We’ve also quantified this overhead in our experiments. The DHRL training procedure resulted in an approximate 3x increase in training time compared to the standard predictive baseline. We’ve added this detail to Appendix C.
>
> ### W1. Dependence on QA-generator.
>
> We view this dependence as a structured mechanism to inject semantic priors from foundation models. We acknowledge this is a limitation in domains where strong priors are unavailable. Crucially, our results (Table 3) show that a fixed generator bootstrapped from an off-the-shelf LLM performs nearly as well (within 2-3%) as an adversarially learned one, confirming the practicality of the approach in language-rich domains.
>
> ### W3. Limited evaluation scope.
>
> We focused on recommender systems as they represent a challenging class of POMDPs characterized by high-dimensional partial observability and rich semantics. However, the DHR concept is general. In non-linguistic domains (e.g., robotics), "questions" can be specific predictive queries (e.g., "Distance to obstacle?"). The QA-space effectively becomes a set of General Value Functions (GVFs) [Sutton et al., 2011] or auxiliary tasks.

---

### Author Response · Authors · 2025-11-15
**Comment to all reviewers**

We sincerely appreciate the time and effort the reviewers invested in their thoughtful and constructive feedback. We are encouraged that reviewers found our approach novel (2xd4, 6zd7, YkaA), **principled** (6zd7), **highly interpretable** (2xd4, YkaA, q2La), and supported by **comprehensive experiments** with strong empirical gains (2xd4, 6zd7, q2La).

We have revised the paper to address the concerns raised, significantly improving clarity regarding the training setup (especially the use of future information), strengthening the theoretical foundations (revised proof of Theorem 1), and enhancing the experimental analysis. All updates in the revised version of the paper are marked in red.

## Key Updates and New Ablations:

Crucially, inspired by the reviewers' feedback, we’ve included three new ablation studies and analyses in the revision (Appendix D):

1. Decoupling QA and RL (FT-QA) (R-q2La, R-YkaA): To isolate the impact of the joint optimization vs. the QA structure itself, we test a supervised variant trained only on the QA objective ($\lambda=1$) and compare it against both the full DHR and the standard predictive baseline. The results confirm that joint optimization is essential for maximizing performance.
2. Sufficiency Test (R-q2La): We test if the DHR is sufficient by providing the Decision Agent access to both the DHR and the raw history: $\pi_D(z_t, h_t)$. The improvement over $\pi_D(z_t)$ was negligible (+0.5% reward), strongly suggesting $z_t$ closely approximates a sufficient statistic.
3. Lambda Sensitivity (R-YkaA): We add a sensitivity analysis for $\lambda$, showing robust performance across the optimal range (0.01-0.1).

We’ve also added 95% confidence intervals to Tables 6 & 7 (Appendix D) confirming statistical significance, and quantified the training overhead (Appendix C).

---

> ### Author Response · Authors · 2025-11-24
>
> Dear Reviewers,
>
> We would like to thank you again for your insightful and constructive feedback, which has significantly helped us improve our paper. We understand that this is a busy period. As the discussion phase continues, we wanted to gently follow up and ensure you have had the opportunity to review our detailed rebuttal and our revised paper (with changes marked in red). We believe we have thoroughly addressed your key concerns, including: (1) clarification on the use of future information (Re q2La, 6zd7), (2) a new ablation FT-QA (Re q2La, YkaA), (3) new ablation for sufficiency test (re q2La), and (4) our theoretical revision and significance (re 2xd4, q2La).
>
> We believe these clarifications and new experiments robustly address the major concerns raised and strongly support the contributions of our work. Please let us know if you have any remaining questions. We are happy to engage in further discussion.
>
> Best regards,\
> The Authors

---

### Meta-Review · Area_Chair_quMd · 2025-12-22

**Summary:**

The paper introduces Descriptive History Representations (DHRs) to compress long interaction histories in partially observable settings into sufficient statistics, defined by their ability to answer task-relevant questions about past interactions and future outcomes. It proposes a joint learning framework that balances reward maximization with question-answering capability, and demonstrates the approach on public movie and shopping datasets by producing interpretable textual user profiles that predict preference-driven behavior.

A major point of confusion raised by multiple reviewers is whether the QA generator has access to future user interactions during training in a way that could leak information from the test set, potentially leading to unfair comparisons with baselines. Reviewers also questioned a potential circular dependency between learning a “sufficient” representation and designing “sufficient” questions, and the rebuttal does not provide a convincing resolution to these issues. In addition, reviewers highlighted the method’s reliance on a strong QA generator and raised concerns about its applicability in domains where high-quality QA pairs are scarce; more broadly, the varying effectiveness of foundation models across domains may limit the practical utility of DHRs. Addressing these concerns adequately would require another round of major revision.

**Reviewer Concerns:**

The authors’ rebuttal clarifies some points of confusion and adds additional results. However, several key concerns outlined above remain insufficiently addressed.

**Reviewer Scores:**

Given that the rebuttal only partially addresses the key concerns, reviewers’ scores are most likely to remain unchanged.

---

### Decision · Program_Chairs · 2026-01-26

Reject